

# Synthetic data analysis for early detection of Alzheimer progression through machine learning algorithms

Ana Gabriela Sánchez Reyna[1], Ricardo Mendoza-Gonzalez[1], Huizilopoztli Luna-García[2], José María Celaya Padilla[2], Jorge Alejandro Morgan Benita[2], Carlos H. Espino-Salinas[2], Jorge I. Galván-Tejada[2], David Rondon[3] and Klinge Villalba-Condori[4]

[1] Systems and Computing Department, TecNM/Technological Institute of Aguascalientes, Aguascalientes, Aguascalientes, Mexico
[2] Unidad Académica de Ingeniería Eléctrica, Universidad Autónoma de Zacatecas, Zacatecas, Zacatecas, Mexico
[3] Estudios Generales, Universidad Continental, Arequipa, Peru
[4] Vicerrectorado de Investigación, Universidad Nacional Pedro Henríquez Ureña, Santo Domingo, Dominican Republic

Corresponding authors
Ricardo Mendoza-Gonzalez,
mendozagric@aguascalientes.tecnm.mx
Huizilopoztli
Luna-García, hlugar@uaz.edu.mx

## ABSTRACT

Alzheimer's disease (AD) is a serious neurodegenerative disorder that causes incurable and irreversible neuronal loss and synaptic dysfunction. The progress of this disease is gradual and depending on the stage of its detection, only its progression can be treated, reducing the most aggressive symptoms and the speed of its neurodegenerative progress. This article proposes an early detection model for the diagnosis of AD by performing analyses in Alzheimer's progression patient datasets, provided by the Alzheimer's Disease Neuroimaging Initiative (ADNI), including only neuropsychological assessments and making use of feature selection techniques and machine learning models. The focus of this research is to build an ensemble machine learning model capable of early detection of a patient with Alzheimer's or a cognitive state that leads to it, based on their results in neuropsychological assessments identified as highly relevant for the detection of Alzheimer's. The proposed approach for the detection of AD is presented with the inclusion of the feature selection technique recursive feature elimination (RFE) and the Akaike Information Criterion (AIC), the ensemble model consists of logistic regression (LR), artificial neural networks (ANN), support vector machines (SVM), K-nearest neighbors (KNN) and nearest centroid (Nearcent). The datasets downloaded from ADNI were divided into 13 subsets including: cognitively normal (CN) *vs* subjective memory concern (SMC), CN *vs* early mild cognitive impairment (EMCI), CN *vs* late mild cognitive impairment (LMCI), CN *vs* AD, SMC *vs* EMCI, SMC *vs* LMCI, SMC *vs* AD, EMCI *vs* LMCI, EMCI *vs* AD, LMCI *vs* AD, MCI *vs* AD, CN *vs* AD and CN *vs* MCI. From all the feature results, a custom model was created using RFE, AIC and testing each model. This work presents a customized model for a backend platform to perform one-versus-all analysis and provide a basis for early diagnosis of Alzheimer's at its current stage.

# INTRODUCTION

Alzheimer's disease (AD), is one of the most common neurodegenerative diseases that primarily affects brain tissue (*Masters et al., 2015*; *Ashayeri et al., 2024*), with older adults being the most affected. The global prevalence of dementia exceeds 55 million individuals, a figure projected to escalate to 139 million by 2050, particularly affecting low and middle-income countries. Notably, 60% of those afflicted currently reside in such nations, a proportion of it, is anticipated to surge to 71% by 2050. The onset of a new dementia case presents approximately every 3 s worldwide, yet up to three-quarters of those affected lack a formal diagnosis.

Anxiety regarding dementia is widespread, with nearly 80% of the populace expressing apprehension about its potential onset and a quarter perceiving dementia as inevitable and unpreventable. Misconceptions regarding dementia prevail, as approximately 62% of healthcare practitioners mistakenly stake it as an inherent aspect of aging. Furthermore, a significant portion of caregivers, approximately 35%, admit to concealing a family member's dementia diagnosis.

The toll of caregiving on health is considerable, with over half of caregivers reporting a decline in their own health due to their responsibilities, despite expressing positive sentiments regarding their caregiving roles. These statistics underscore the urgent need for enhanced awareness, support and interventions to address the multifaceted challenges posed by dementia on a global scale (*Long & Benoist, 2023*). It is possible to detect AD through certain biomarkers present in the cerebrospinal fluid and the accumulation of Beta amyloid, as well as the use of neurological images (*Masters et al., 2015*).

Dementia encompasses various neurological conditions characterized by impairments in memory, cognition, behavior and emotion. Early manifestations often include memory loss, challenges in executing familiar tasks, linguistic difficulties and alterations in personality. Despite extensive research efforts, dementia remains devoid of a definitive cure, nevertheless, numerous support mechanisms exist to aid individuals affected by dementia and their caregivers. Remarkably, dementia disregards societal divisions, impacting individuals across all social, economic and ethnic strata.

As AD stands the foremost etiology of dementia, vascular disease, dementia with Lewy bodies and frontotemporal dementia represent significant contributors to this multifaceted syndrome.

By implementing machine learning (ML) and deep learning techniques, which are part of artificial intelligence, the necessary conditions have been generated to be able to process large amounts of information and promote problem solving, the improvement of diagnostic support tools and establish new lines of action based on the results of each analysis.

According to the World Health Organization (WHO), the global population aged 60 and above reached approximately one billion individuals in 2020, with projections indicating a

doubling to 2.1 billion by the foreseeable future. Notably, the demographic of individuals aged 80 and older is anticipated to undergo a threefold increase during this period, reaching a staggering 426 million individuals.

The economic burden associated with dementia is poised to escalate significantly, WHO forecasts suggest that the annual costs linked to dementia will more than double from $1.3 trillion in 2019 to an estimated $2.8 trillion by 2030. This public challenge underscores the imperative need for proactive measures in healthcare policy, research and caregiving infrastructure to address the evolving needs of aging populations and mitigate the socioeconomic ramifications of dementia on a global scale (_Long & Benoist, 2023_).

Assessment of AD progression encompasses a multifaceted approach, leveraging specialized clinical data amalgamated from comprehensive clinical histories, neuropsychological evaluations and diverse clinical assessments like blood assays and electroencephalography. This comprehensive array of studies serves to discern risk factors and distinguish symptoms akin to dementia. Through integration into machine learning frameworks, this data enables predictive evaluation criteria, facilitating tailored action plans or additional investigations based on the patient's current condition or accurate exclusion of the ailment. Simultaneously, clinical features can be simulated, marking a pivotal stride towards personalized medicine.

## Related works

Currently, there are different works related to the detection of AD through the implementation of ML algorithms, each approach presents interesting methodological proposals that address the problem from different perspectives seeking to understand more about this type of disease that affects a large part of the population and thus improve their quality of life, there are some comparisons in related work that can be pointed in Table 1. The works proposed in the state of the art are shown below.

### _Machine learning in Alzheimer detection_

The work of _Fisher et al. (2019)_ underscores the utility of ML models, exemplified by their Conditionally Constrained Boltzmann Machine, in simulating intricate patient trajectories. However, these trajectories necessitate vigilant monitoring to gauge disease progression in individuals with cognitive impairment or AD, benchmarked against cognitively normal (CN) individuals. This facilitates model training for personalized disease prognosis. Notably, the adoption of synthetic or simulated data, derived from actual datasets, emerges as an increasingly pertinent practice. Given the exigency for close patient monitoring and the likelihood of incomplete data due to patient non-compliance with rigorous testing, synthetic data aids in circumventing analytical challenges associated with data sparsity.

The identification of individuals at high risk of transitioning from mild cognitive impairment (MCI) to AD demands meticulous surveillance, targeted investigations and personalized treatment interventions. Given the economic burden associated with early detection and continual monitoring, recent efforts have turned towards leveraging various ML algorithms for the detection of AD progression (_El-Sappagh et al., 2021_). However, most studies in this domain rely solely on neuroimaging data obtained during baseline visits.

**Table 1  Comparison of machine learning models for Alzheimer's disease prediction.**

| Title | Advantages | Disadvantages | Gaps |
|---|---|---|---|
| A Comprehensive Machine-Learning Model Applied to Magnetic Resonance Imaging (MRI) to Predict Alzheimer's Disease (AD) in Older Subjects (*Battineni et al., 2020*) | High Accuracy, Diverse Models, Feature Selection, Ensemble Modeling, Empirical Validation and Relevance | Limited Dataset, Age Restriction, Model Complexity and Single Data Source | Longitudinal Validation, External Validation, Feature Interpretability, Real-World Application and Comparative Analysis |
| Magnetic Resonance imaging biomarkers for the early diagnosis of Alzheimer's disease: a machine learning approach (*Salvatore et al., 2015*) | Identification of Critical Brain Regions, Specific Focus on MCI Conversion, Use of Optimized Machine Learning Algorithm, Application to Clinical Practice and Encouraging Accuracy Rates | Moderate Accuracy, Limited Generalizability, Complexity of Implementation and Potential Overfitting | Longitudinal Validation, External Validation, Integration with Other Biomarkers, Detailed Feature Analysis, Impact on Clinical Decision-Making |
| Evaluating the reliability of neurocognitive biomarkers of neurodegenerative diseases across countries: A machine learning approach (*Bachli et al., 2020*) | High Accuracy, Multimodal Data Integration, Cross-Country Validation, Identification of Relevant Features and Clinical Applicability | Complexity in Implementation, Limited Cohort Size, Focus on Specific Conditions and Potential Overfitting | Longitudinal Validation, Integration with Other Biomarkers, Detailed Feature Analysis, Impact on Clinical Workflow and Validation in Broader Populations |
| Enhancing Early Dementia Detection: A Machine Learning Approach Leveraging Cognitive and Neuroimaging Features for Optimal Predictive Performance (*Irfan, Shahrestani & Elkhodr, 2023*) | Simplicity of Cognitive Tests, Robust Performance of AdaBoost Ensemble Model, Potential for Early Detection, Combination of Cognitive and Neuroimaging Features and Machine Learning Application | Limited Patient Records, Feature Identification Uncertainty, Comparative Performance, Lack of Longitudinal Data and Interpretability of Models | Generalizability Across Diverse Population, Integration with Clinical Workflow, Validation with Larger Datasets and Impact on Patient Outcomes |

**Table 1** (*continued*)

| Title | Advantages | Disadvantages | Gaps |
|---|---|---|---|
| Assessment for Alzheimer's Disease Advancement Using Classification Models with Rules (*Thabtah & Peebles, 2023*) | Early Intervention and Treatment Planning, Non-invasive and Accessible, Efficiency and Affordability, Comprehensive Cognitive Assessment, High Performance and Interpretability | Dependence on Specific Datasets, Generalizability, Limited Cognitive Domains, Initial Stage Focus and Resource Constraints | Validation Across Diverse Populations, Longitudinal Studies, Integration with Clinical Workflows, Comparison with Latest Techniques and Impact on Patient Outcomes |

One of the most studied methodologies in the field of early detection of AD is image analysis as proposed by *Kumari, Nigam & Pushkar (2020)*, where a total of 200 brain magnetic resonance imaging (MRI) images, 100 images for testing and 100 images for training are considered to generate a ML model to detect AD in its early stages. The model applied a Gaussian filter to remove unwanted noise, an Otsu threshold for image segmentation, a Prewitt edge detection approach, Gray level co-occurrence matrix (GLCM) for feature extraction, fuzzy c-means (FCM) for clustering and finally a convolutional neural network (CNN) for image classification. The model showed a pressure of 90.24% and a sensitivity of 85.53%. Likewise (*Battineni et al., 2021*), with demographic information, MRI and pre-existing patient conditions can help improve classifier performance. In the study, a supervised learning classifier-based framework was proposed in the categorization of subjects with dementia based on longitudinal brain MRI features. The classification results of the proposed algorithm outperforms other models with 97.58% of accuracy. Also *Dashtipour et al. (2021)*, proposed a framework based on ML and deep learning methods to detect AD from raw data from MRI scans, evaluated the performance of different algorithms for comparison purposes. Experimental results indicate that the Bidirectional Long Short Term Memory (BiLSTM) network outperforms ML methods with a detection accuracy of 91.28% (*Alroobaea et al., 2021*). They used brain data from the Alzheimer's Disease Neuroimaging Initiative (ADNI) and Open Access Imaging Studies (OASIS). They applied common Alzheimer's ML techniques, such as logistic regression (LR), support vector machines (SVM), random forests, linear discriminant analysis and more. The best accuracy values given by ML classifiers are 99.43% and 99.10% given by LR and SVM using the ADNI dataset.

Nonetheless, AD, being a complex chronic condition, necessitates the expertise of specialized medical professionals to meticulously analyze patients' histories to establish a diagnosis of progression. Unfortunately, the availability of analyzed neuroimaging data remains limited, particularly in developing countries, primarily due to its high cost.

To replicate and extend such analyses with prediction horizons typically exceeding 1 year, diverse ML models are being developed and compared. These models are designed to accommodate the intricate structure of the data, extracting patterns from varied collections of informative time-series features. These may include patient comorbidities, cognitive

assessments, medication histories and even demographic information (*El-Sappagh et al., 2021*).

ML tools have emerged as powerful instruments for modeling intricate relationships among diverse clinical variables, surpassing human capabilities in discerning complex patterns within clinical data. When applied to blind testing—data unseen during model training—the output of trained ML models furnishes insights that can enhance clinical decision-making in diagnosis. The evolution of computational resources has propelled researchers from simplistic ML algorithms like regression toward more sophisticated Deep Learning models.

### Deep learning in Alzheimer detection

Prominent ML models encompass regression, SVM, decision trees, Bayesian networks, artificial neural networks (ANN) and natural language processing (NLP). ANN encompass classifiers such as multilayer perceptrons, along with deep learning models like convolutional ANN and autoencoders. ML models are categorized based on learning algorithms as supervised, unsupervised, or semi-supervised. In supervised learning, algorithms are trained on a dataset annotated with gold standard labels. Unsupervised learning models discern features and patterns from unlabeled data, organizing data points into distinct classes (*Kumar et al., 2021*). For early detection efforts to be effective, accessible and multimodal diagnostic instruments are indispensable. These tools should ideally be provided by international organizations or governmental bodies either free of charge or at minimal cost, ensuring broad accessibility and equitable healthcare provision.

While diagnostic methods based on imaging exhibit significant precision and certainty in predicting the advancement of AD, the challenge lies in the overwhelming burden placed on specialized medical personnel required to interpret these images and provide diagnoses, particularly given the exponential increase in the number of patients compared to the limited number of specialists available. MRI offers detailed analysis of specific brain tissues, enabling accurate diagnoses when segmented. However, MRI teams, due to their high infrastructure and service costs, cater to only a fraction of the population, despite the existence of ML models capable of analyzing images with precision comparable to that of specialized doctors.

This scenario often implies that individuals already exhibit advanced deterioration by the time they undergo imaging, as routine check-ups typically do not include neuropsychological assessments or assessments specifically targeting AD. Consequently, many people do not undergo advanced studies, resulting in missed opportunities for early diagnosis. Sparse and unbalanced labeled datasets pose challenges for ML models, necessitating synthetic oversampling techniques. To address this, models utilizing synthetic data must go through validation by expert personnel or be trained on balanced datasets for reliable comparison, such as those employed by VGG16 and EfficientNet (*Mujahid et al., 2023*). These considerations underscore the importance of enhancing access to advanced diagnostic technologies and ensuring the availability of balanced datasets to facilitate early and accurate diagnosis of AD.

Current approaches integrate medical history, neuropsychological testing and MRI data, yet efficacy remains inconsistent due to limitations in sensitivity and precision. The research efforts focus on image processing utilizing CNN algorithms to discern specific features of AD from MRI images. These studies typically delineate AD into four stages, employing CNN-based models to generate high-resolution disease probability maps. These maps are then fed into a multilayer perceptron, yielding precise and intuitive visualizations of individual AD risk. A recent case is presented by *Yashodhar & Kini (2024)*, who proposes considering a CNN and SVM as a basis for evaluating different methodologies and obtaining the best possible results. The study demonstrates the ability of tabular data alone to obtain assertive diagnoses for early detection of AD that support different methodologies based on this approach.

### Imbalance data for ML models

Another significant challenge lies in addressing class imbalance within datasets, necessitating even distribution of samples across classes. Although MRI datasets from platforms like Kaggle are utilized, they often suffer from pronounced class imbalance issues. Notably, *Murugan et al. (2021)* leverage the Kaggle platform to implement a CNN named DEMentia NETwork (DEMNET) for dementia stage detection from MRI data providing a useful tool for AD early diagnosis (*Murugan et al., 2021*). While their results utilizing the Kaggle dataset are noteworthy, they validate their findings using datasets from ADNI, which serves as a crucial benchmark in AD diagnosis. Despite that, several methods combining resampling techniques with classification approaches have been proposed to address imbalanced data (*Ahmad et al., 2024*), achieved successful validation by applying different methods, including a synthetic data generation process to solve data imbalance problems that usually cause a significant decrease in the performance of ML models. Also, studies such as that of *Öter & Doğan (2024)*, address challenges of unbalanced datasets in AD classification using techniques such as SMOTE, ADASYN and weight balancing. Their proposal succeeded in demonstrating that the ADASYN method performed superior, achieving the highest accuracy scores. The study emphasizes the importance of addressing class imbalances in Alzheimer's datasets to create more accurate and reliable models.

The utilization of image processing techniques in medical applications confronts several noteworthy limitations warranting attention. One of them is the size of datasets employed for training and evaluation tends to be relatively restricted. Constructing such datasets necessitates substantial investments of time and financial resources, potentially impeding the development of models trained on larger and more diverse datasets. Ensuring the representativeness of these datasets demands further research to assess model performance across broader spectra but the availability of computational resources poses constraints on the scale and complexity of experiments. The limitations in computational resources may restrict the extent to which models can be trained or the complexity of algorithms that can be employed. There is a pressing need to address computational efficiency, particularly for real-time processing applications. The streamlining algorithms to optimize computational resources are essential for facilitating timely diagnosis and interventions, nevertheless, ethical considerations loom large, particularly regarding the privacy and

security of sensitive medical data. Safeguarding patient privacy and ensuring data security must be paramount in the development and deployment of image processing techniques in clinical settings. Adherence to ethical guidelines and robust data protection measures are imperative to foster trust and uphold patient confidentiality in medical research and practice (*Rana et al., 2023*).

Unlike the predominant focus on imaging studies in existing literature, neuropsychological assessments have received scant attention over the years. This is primarily due to the broad and often complex nature of validated instruments, making them challenging to administer to large groups of individuals. To address this issue, a preliminary selection of the most pertinent items from neuropsychological assessments is imperative to construct an optimal questionnaire capable of detecting varying degrees of dementia. Typically, these assessments are developed collaboratively with neuropsychologists who design the instrument for implementation. Subsequently, an algorithm is proposed to select relevant features, thereby reducing computational costs and enhancing interpretability by isolating elements directly correlated with a patient's mental condition. To retain features with superior predictive performance, expedite and economize predictor generation, diminish dimensionality and mitigate the risk of overfitting during the training phase, information gain feature selection algorithms are employed. Information gain, rooted in information theory, is a widely utilized method in data mining (*Zhu et al., 2020*). It quantifies the amount of information a feature can contribute to the classification model. Features with higher information gain values for a particular class harbor greater classification information pertinent to that class. By employing information gain-based feature selection algorithms, features with lower scores are sequentially discarded, while those deemed informative are retained for model input. This iterative process facilitates the identification of a feature set with a reduced number of elements, minimizing any accompanying reduction in classification accuracy.

Predicting the progression from MCI to AD represents a highly intricate endeavor. Neuropsychological evaluation plays a crucial role in identifying individuals with mild cognitive impairment (MCI) at an early stage, offering the potential for a more controlled and manageable disease trajectory before advancing to dementia. However, the multitude of neuropsychological tests (NPTs) conducted in clinical settings, coupled with the limited number of training examples, presents a formidable challenge for ML in constructing prognostic models. Each NPT comprises numerous elements and undergoes rigorous validation by medical or ethics boards, further complicating the ML learning process. To mitigate these challenges, it is often proposed to streamline NPTs by reducing the number of elements or utilizing ML to identify the most informative features relevant to the analyzed state. These subsets of NPTs, from which prognostic models are trained, must not only demonstrate predictive efficacy but also exhibit stability and facilitate the development of generalizable and interpretable models. Efforts to distill NPTs into concise yet informative sets of features enable the construction of robust ML models capable of accurately predicting disease progression while ensuring clinical relevance and interpretability (*Pereira et al., 2018*).

### Neuropsychological evaluations

The integration of clinical and anthropometric data alongside neuropsychological assessments has emerged as a promising approach for early dementia diagnosis, particularly in dementia support centers. The ML techniques are leveraged to analyze such datasets, which typically encompass patient demographics such as sex, age and education, along with results from neuropsychological assessments like the Mini Mental State Examination in the Korean version of the CERAD Evaluation Package (MMSE-KC), commonly employed as a screening tool for dementia (*So et al., 2017*). *Gaeta et al. (2024)*, propose to develop a multimodal ML model as an early diagnostic tool to accurately predict cases of AD using a specific set of noninvasive variables as possible indicators of neuropathology of the disease. It emphasizes that the importance of specific features of the quantitative polysomnography (PSG) signal, such as electroencephalogram (EEG) asymmetry, Lempel's Ziv and sample entropy of thoracic stress signals, as reliable markers to predict neurodegeneration. The feasibility of the approach underscores its potential contribution to early diagnosis through comorbidities, sociodemographic information and exploration of its relationships with sleep patterns.

In these datasets, the initial classification task often involves distinguishing between normal and abnormal cases based on the available clinical and neuropsychological data. Following this initial filtering step, ML models are employed to further classify cases into specific categories, such as dementia or cognitive impairment. By harnessing the combined power of clinical, anthropometric and neuropsychological data, ML techniques facilitate the development of robust diagnostic models capable of accurately identifying individuals at risk of dementia or cognitive decline. This integrated approach holds promise for enhancing early detection and intervention efforts in dementia care settings.

Several neuropsychological evaluations are available to assess the cognitive state of individuals and determine the extent of cognitive and behavioral decline. Among the most commonly used assessments is the Mini-Mental State Examination (MMSE) (*Folstein, Folstein & McHugh, 1975*), which comprises a standardized set of questions used globally to measure cognitive impairment. The subject's level of cognitive impairment is reflected in the score obtained on this evaluation calculated by summing the number of correctly answered questions; a lower score indicates greater cognitive impairment. The work presented by *Prabhakaran et al. (2024)* demonstrated that combining multiple sociodemographic factors with neuropsychological test measures provides a robust prediction of conversion to MCI on a patient-by-patient basis. Current approaches provide a basis for integrating longitudinal bases for predicting the onset of pre-clinical dementia or MCI to other non-Alzheimer's disease dementias. Determination and validation of key predictive measures may enable clinicians to monitor patients' health more efficiently.

Another frequently utilized neuropsychological assessment is the cognitive subscale of the Alzheimer's Disease Assessment Scale (ADAS), consisting of 11 items (ADAS-Cog 11) (*Rosen, Mohs & Davis, 1984*), with a variant containing two additional items (ADAS-Cog 13) (*Mohs et al., 1997*). These subscales are components of the ADAS designed to evaluate cognitive dysfunction and the severity of cognitive symptoms associated with dementia. There are assessments for global dementia scaling that clinically evaluate and categorize its

**Table 2  Neuropsychological assessments for AD diagnosis (scores).**

| Assessments | Score range | Score | Stages of cognitive function |
|---|---|---|---|
| MMSE | 0–30 | 24–30 | Normal cognitive |
| | | 19–23 | Mild dementia |
| | | 10–18 | Moderate dementia |
| | | <9 | Severe dementia |
| ADAS-Cog 11 | 0–70 | | Higher scores suggest greater severity of the cognitive symptoms of dementia |
| ADAS-Cog 13 | 0–85 | | Higher scores suggest greater severity of the cognitive symptoms of dementia |
| CDGLOBAL | 0–3 | 0 | No dementia |
| | | 0.5 | Questionable dementia |
| | | 1 | MCI |
| | | 2 | Moderate cognitive impairment |
| | | 3 | Severe cognitive impairment |

progression and severity, such as the Clinical Dementia Rating (CDR), which is a staging tool used to classify the severity of dementia (*Morris, 1991*). The CDR generates a global score (CDGLOBAL) that determines the stage of dementia and this score is calculated using a specific algorithm. The other score is a box sum score (CDRSB) that measures the severity of dementia and it is calculated by summing the scores of each domain box (see Table 2). Therefore, these evaluations play a pivotal role in the diagnosis of AD.

## METHODS

According to the stages of the processes presented in the research of *Sánchez-Reyna et al. (2021)* and *Morgan-Benita et al. (2022)*, the methodology proposed for this study also consists of six stages, adapting each individual process to the experimentation carried out in this study, as shown in the Fig. 1. Firstly, in Stage 1 (Fig. 1A), a comprehensive description of the datasets utilized in the study is provided. Following this, Stage 2 (Fig. 1B) involves the creation of the dataset of interest by selecting subjects based on predefined inclusion criteria. Additionally, this stage encompasses the verification and treatment of empty fields within the dataset, as well as data transformation procedures as necessary. In Stage 3 (Fig. 1C), the dataset is divided into thirteen different groups, which are then segmented into two experimental phases. In Phase 1, ten datasets are used due to the incorporation of five cognitive states for subject diagnosis. These ten datasets are mentioned below: CN *vs* subjective memory concern (SMC), CN *vs* early mild cognitive impairment (EMCI), CN *vs* late mild cognitive impairment (LMCI), CN *vs* AD, SMC *vs* EMCI, SMC *vs* LMCI, SMC *vs* AD, EMCI *vs* LMCI, EMCI *vs* AD and LMCI *vs* AD. On the other hand, Phase 2 involves three datasets, as a reorganization of the diagnoses is carried out in accordance with the ADNI criteria, resulting in the use of three cognitive states for subject diagnosis. Phase 2, three datasets are delineated as follows: CN *vs* MCI, CN *vs* AD and MCI *vs* AD. Furthermore, an additional dataset is generated alongside the original one. The original dataset is termed an "Original" dataset and a "Synthetic" dataset is created from it. Both datasets are utilized

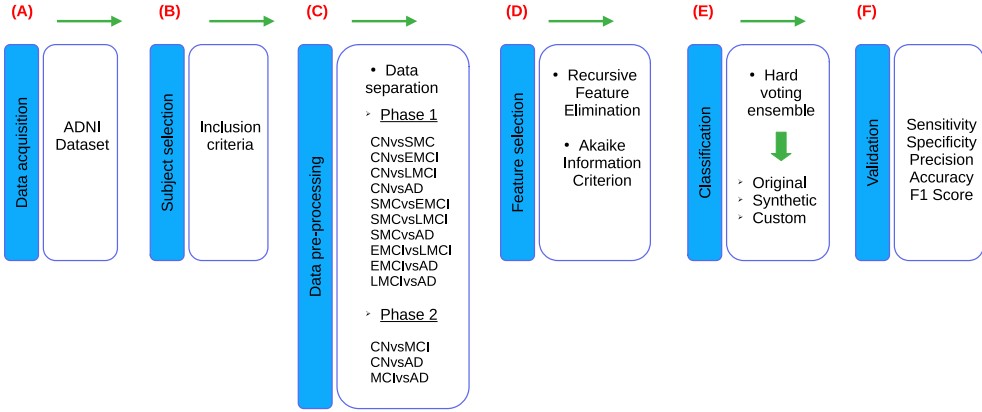

**Figure 1** **Flowchart of the proposed methodology, based on** *Sánchez-Reyna et al. (2021)* **and** *Morgan-Benita et al. (2022)*.

for each of the classifications proposed in this stage, resulting in a total of twenty-six datasets for developing models in the classifier ensemble. Subsequently, in Stage 4 (Fig. 1D), feature selection is conducted employing recursive feature elimination (RFE) and Akaike Information Criterion (AIC) techniques to identify the most relevant neuropsychological assessments for analysis. In stage 5 (Fig. 1E), a set of classifiers is used to generate ML models to evaluate the set of neuropsychological assessments obtained from the previous stage. This stage is designed to evaluate the predictive ability of selected neuropsychological assessments to distinguish between various cognitive states. This process is performed on both Original and Synthetic datasets. Moreover, once the best set of neuropsychological assessments in the Synthetic dataset was identified, these neuropsychological assessments were taken for "Custom model" building, with the goal of generating a smaller ML model, allowing to reduce the number of neuropsychological assessments, so that the selected neuropsychological assessments can distinguish between various cognitive states. Finally, in Stage 6 (Fig. 1F), the performance of the developed models is validated using various metrics including sensitivity, specificity, precision, accuracy and F1-score. These metrics are employed to evaluate the robustness and efficacy of the models in accurately predicting disease progression and classification.

## ADNI database

The data used in this study were obtained from the ADNI database (adni.loni.usc.edu; accessed on 15 October 2023). The ADNI was launched in 2003 as a public–private partnership, led by the Principal Investigator Michael W.Weiner, MD. The primary goal of ADNI has been to test whether serial MRI, PET, other biological markers and clinical and neuropsychological assessments can be combined to measure the progression of MCI and early AD (*ADNI, 2003*). For up-to-date information, see http://www.adni-info.org; accessed on 15 October 2023.

**Table 3  Inclusion criteria.**

| Inclusion criteria |
| --- |
| 1. Patients should be assigned baseline (BL) and subsequent visit codes. |
| 2. Patient ages must fall within the range of 53 to 95 years at the time of enrollment. |
| 3. Only patients identifying as Hispanic/Latino should be included. |
| 4. No differentiation should be made based on sex, education, race, or marital status. |
| 5. Datasets should exclusively consist of neuropsychological assessments. |
| 6. The datasets should differentiate between CN, SMC, EMCI, LMCI, MCI and AD. |
| 7. Complete data for each feature of each subject is necessary. |

## Inclusion criteria

The dataset known as ADNIMERGE, consists of 16,431 observations and 116 features (*The ADNI team, 2021*). To cleanse the ADNIMERGE dataset, a set of inclusion criteria was implemented, as outlined in Table 3. Firstly, a filter was applied to the VISCODE feature to exclude observations with missing values (NA) and subsequently, this column was removed from the dataset. Additionally, all patients within the age range of 53 to 95 years were retained. From the PTETHCAT column, only records corresponding to patients identified as Hispanic/Latino were retained to ensure the inclusion of individuals from Latinoamerican backgrounds. Subsequently, this column was removed as it became redundant given that all patients were established to be Hispanic/Latino.

The dataset did not discriminate based on sex (PTGENDER), education level (PTEDUCAT), race (PTRACCAT), or marital status (PTMARRY). Clinic data was deemed irrelevant for the study and thus removed from the dataset. The dataset classified patients into different cognitive states including cognitively normal (CN), subjective memory concern (SMC), early mild cognitive impairment (EMCI), late mild cognitive impairment (LMCI), mild cognitive impairment (MCI) and Alzheimer's disease (AD).

Finally, all features and observations with NA were eliminated to ensure the consistency of the included data. Following data purification, the dataset was reduced to 323 observations without any missing values, down from the initial 552 observations. This meticulous cleansing process aimed to enhance the reliability and integrity of the dataset for subsequent analysis.

Qualitative variables are transformed into quantitative ones with a nominal scale.

- NC = 0; SMC = 1; EMCI = 2; LMCI = 3; AD = 4.
- Female = 0; Male = 1.
- Nevermarried = 0; Married = 1; Divorced = 2; Widowed = 3.

After applying the inclusion criteria, a dataset with 14 features and 323 observations was obtained. This new dataset is called the "Original dataset". Table 4 shows the resulting features.

## Dataset separation

The dataset was partitioned into 13 distinct cases, each representing different comparisons between cognitive states based on the DX.bl feature, which designates the diagnosis

**Table 4** Features and description of the "original dataset" after the application of the inclusion criteria to the ADNI data.

| Features | Description |
|---|---|
| AGE | Age of the participant |
| PTGENDER | Gender of the participant |
| PTMARRY | Marital status of the participant |
| CDRSB | Clinical Dementia Rating Scale Sum of Boxes score |
| ADAS11 | Alzheimer's Disease Assessment Scale, 11-item version |
| ADAS13 | Alzheimer's Disease Assessment Scale, 13-item version |
| ADASQ4 | Alzheimer's Disease Assessment Scale, Question 4 score |
| MMSE | Mini-Mental State Examination score |
| RAVLT.immediate | Immediate recall score from the Rey Auditory Verbal Learning Test |
| RAVLT.learning | Learning score from the Rey Auditory Verbal Learning Test |
| RAVLT.forgetting | Forgetting score from the Rey Auditory Verbal Learning Test |
| LDELTOTAL | Total score from the Longitudinal Data Entry List |
| TRABSCOR | Total score from the Test of Recent Abstraction |
| FAQ | Functional Activities Questionnaire score |

corresponding to the patient's cognitive state. These cases provide insights into the subtle yet discernible differences observed in neuropsychological test results among patients classified under various cognitive states according to the ADNI.

In the initial phase (phase 1) of the dataset analysis, patients were diagnosed with one of five cognitive statuses: CN, SMC, EMCI, LMCI or AD.

In a subsequent phase (phase 2) of the analysis, these diagnoses were reorganized based on the available ADNI information (*ADNI, 2003*). CN and SMC were combined into a single group labeled CN, while EMCI and LMCI were grouped together as MCI. AD retained its original diagnosis.

1. CN *vs.* SMC (phase 1): This comparison involves observations where patients exhibit a CN and those with SMC. By examining the differences between these two groups, we can extract insights into the correlation between neuropsychological test results and the subjective experience of memory concerns in patients.

2. CN *vs.* EMCI (phase 1): This comparison aims to identify the most suitable neuropsychological assessments for detecting EMCI among patients who are initially considered CN. The selected assessments can serve as a basis for developing instruments to detect the early stages of EMCI.

3. CN *vs.* LMCI (phase 1): Observations in this comparison highlight the distinctions between patients classified as CN and those with LMCI. This analysis provides valuable insights into the progression of cognitive decline and identifies individuals who may be at risk of developing AD.

4. CN *vs.* AD (phase 1): This comparison contrasts CN individuals with those diagnosed with AD. By examining the differences between these two groups, we gain a better

understanding of the distinctive features associated with severe cognitive impairment and the transition from normal cognition to AD.

5. SMC *vs.* EMCI (phase 1): The observations highlight subtle differences between individuals with SMC and those with EMCI. While the cognitive state of individuals in the EMCI group may exhibit slight deterioration compared to those with SMC, the progression from SMC to EMCI is discernible. This comparison aids in understanding the continuum of cognitive decline and provides insights into the early stages of cognitive impairment.

6. SMC *vs.* LMCI (phase 1): In this comparison portrays a clearer progression in cognitive decline. Individuals classified as having SMC demonstrate a perceptible deterioration in cognitive function compared to those with LMCI. This comparison underscores the advancing stages of cognitive decline and highlights the transition from SMC to more pronounced impairment.

7. SMC *vs.* AD (phase 1): This comparison elucidates a clear difference between cognitive states. Individuals with SMC exhibit distinct cognitive features compared to those diagnosed with AD, indicating a significant progression in cognitive decline from subjective concerns to severe impairment associated with AD. This comparison underscores the importance of early detection and intervention for individuals exhibiting SMC, as they may be at increased risk of developing AD.

8. EMCI *vs.* LMCI (phase 1): In this comparison the observations reveal a notable distinction between individuals with EMCI and those diagnosed with LMCI. This comparison sheds light on the progression of cognitive decline from early stages of impairment to more severe manifestations associated with LMCI, providing valuable insights into the diagnostic criteria and trajectory of the disease.

9. EMCI *vs.* AD (phase 1): In this comparison the observations reveal a notable distinction between individuals with EMCI and those diagnosed with AD. This comparison sheds light on the progression of cognitive decline from early stages of impairment to more severe manifestations associated with AD, providing valuable insights into the diagnostic criteria and trajectory of the disease.

10. LMCI *vs.* AD (phase 1): This comparison showcases a discernible progression in cognitive decline, with individuals classified as having LMCI exhibiting more pronounced impairment compared to those with AD. This comparison underscores the advancing stages of cognitive decline and highlights the transition from mild impairment to severe cognitive deficits feature of AD.

11. CN *vs.* MCI (phase 2): This comparison, observations reveal differences between CN individuals and those with MCI. This comparison provides insights into the early stages of cognitive decline and underscores the importance of identifying individuals at increased risk of developing AD for early intervention and management strategies.

12. CN *vs.* AD (phase 2): This comparison clearly shows disparities emerge between CN individuals and those diagnosed with AD. This comparison highlights the stark contrast in cognitive function between healthy individuals and those experiencing severe cognitive impairment, underscoring the importance of early detection and intervention in the management of AD.

13. MCI *vs.* AD (phase 2): This comparison, observations demonstrate significant differences between individuals with MCI and those diagnosed with AD. This comparison underscores the importance of distinguishing between different stages of cognitive impairment and highlights the need for early detection and intervention to potentially delay progression to AD.

The dataset was divided into 13 subdatasets and analyzed in two phases to thoroughly investigate the differences in cognitive states based on the DX.bl feature, which indicates the diagnosis related to the patient's cognitive state. These subdatasets provide detailed insights into the variations observed in neuropsychological test results among patients classified under different cognitive states as per the ADNI.

In the initial phase (phase 1) of the analysis, patients were diagnosed with one of five cognitive statuses: CN, SMC, EMCI, LMCI, or AD. This phase involved comparisons such as CN *vs.* SMC, CN *vs.* EMCI, CN *vs.* LMCI, CN *vs.* AD, SMC *vs.* EMCI, SMC *vs.* LMCI, SMC *vs.* AD, EMCI *vs.* LMCI, EMCI *vs.* AD, and LMCI *vs.* AD. Each of these comparisons aimed to highlight subtle yet significant differences in cognitive function and the progression of cognitive decline from normal cognition or mild concerns to severe impairment.

In the subsequent phase (phase 2), the diagnoses were reorganized based on additional information available from ADNI. CN and SMC were combined into a single group labeled CN, while EMCI and LMCI were grouped together as MCI. AD retained its original diagnosis. This phase included comparisons such as CN *vs.* MCI, CN *vs.* AD, and MCI *vs.* AD, which provided insights into broader categorizations of cognitive decline and helped in understanding the transitions from normal cognition to mild impairment and from mild impairment to severe cognitive deficits.

By organizing the dataset into these 13 subdatasets across two phases, the study was able to systematically explore and document the continuum of cognitive decline, providing valuable information for early detection, diagnosis and potential intervention strategies in the management of cognitive impairments and AD. This structured approach ensures that the analysis is comprehensive and captures the nuanced differences in cognitive states, ultimately contributing to a better understanding of the progression of cognitive decline and aiding in the development of targeted treatments and diagnostic tools.

## Synthetic dataset generation

The datasets used in this study are referred the "Original", which serve as the basis for the creation of the "Synthetic", according to each of the comparisons made in the two phases of this study. The Synthetic dataset is generated from the Original dataset through advanced data augmentation techniques, specifically utilizing the AdaBoost algorithm, to enhance the diversity and volume of data available for analysis. AdaBoost, or Adaptive Boosting, is a powerful ensemble technique that creates multiple weak classifiers and combines them to form a strong classifier, thus improving the model's performance and reducing errors. By leveraging AdaBoost for data augmentation, the Synthetic dataset captures a broader range of variations and patterns that may not be present in the Original dataset alone (*Thanathamathee & Lursinsap, 2013*). This process is carefully designed to ensure that the

synthetic data generation accurately captures the underlying distribution and patterns present in authentic patient data without introducing artificial biases or distortions. Both the Original and Synthetic datasets are employed in the classification tasks proposed in this stage of the research. This approach ensures that the models developed can be rigorously tested and validated across different data scenarios. The use of both types of datasets results in a total of twenty-six distinct datasets, each tailored to develop and refine the models within the classifier ensemble framework. By leveraging both Original and Synthetic datasets, the study aims to improve the robustness, generalizability and accuracy of the ML models, ultimately providing more reliable predictions and insights. This comprehensive approach not only mitigates the limitations of working with small sample sizes but also enables the exploration of the models' performance under varied conditions, thereby enhancing the overall quality and applicability of the research findings.

## Cross validation

In all the ML model implementations a 10-fold cross-validation was used. 10-fold cross-validation is a robust and widely used method for evaluating ML models (*Yadav & Shukla, 2016*). The process involves partitioning the dataset into ten subsets, or "folds". Each fold is used as a validation set while the remaining nine folds are used for training. This process is repeated ten times, with each fold serving as the validation set once. The results from each iteration are then averaged to produce a single estimation of model performance. Some of the benefits using this technique are the reduction of overfitting, which helps to ensure that the model generalizes well to unseen data, thereby reducing the risk of overfitting. Other is a comprehensive performance evaluation that ensures the performance evaluation is based on the entire dataset, providing a more comprehensive and reliable metric optimizing the bias–variance tradeoff and provides a robust estimate of model performance while maintaining computational efficiency, making it suitable for datasets of varying sizes.

## Feature selection

The RFE is a feature selection algorithm widely utilized in ML to identify the most relevant features for predictive modeling. This iterative approach aims to enhance model performance by systematically removing less informative features from the dataset.

The procedure begins by training a ML model on the complete feature set. Subsequently, the algorithm ranks the features based on their importance or contribution to the model's performance. The least important feature(s) are then removed from the dataset and the model is retrained on the reduced feature set (*Divya, Shantha Selva Kumari & Initiative, 2021*). This process is repeated iteratively until a predefined number of features or a stopping criterion is met.

The output of the RFE algorithm is a subset of features that yield the best performance on the model. By iteratively evaluating and refining the feature set, RFE helps improve model interpretability, reduce overfitting and enhance predictive accuracy. This iterative process enables the identification of the most informative features, thereby facilitating more efficient and effective predictive modeling in ML tasks (*Carlos, Jesús & De Valladolid Escuela Técnica Superior De Ingenieros De Telecomunicación, 2023*).

The RFE algorithm is utilized to identify the optimal set of neuropsychological assessments that can effectively assess the most significant changes in the cognitive state of patients, in both phases of the study described in the previous section 'Dataset separation'.

The number of features in the final result is determined by evaluating the AIC and accuracy of the feature set provided by RFE in each iteration and comparing each set with the previously evaluated one. This use of AIC ensures smaller models, while accuracy guarantees that only the highest-performing model with each set of features is selected. This modification introduces a novel approach to RFE that eliminates the need for a predefined number of features to conclude the elimination process.

The mathematical calculations for RFE are presented in a series of formulas:

1. **Initial feature set:**

   $$\mathcal{F} = \{f_1, f_2, \ldots, f_n\}$$

   where $\mathcal{F}$ is the initial set of features.

2. **Train the model**: Train a model $\mathcal{M}$ on the current set of features $\mathcal{F}$.

3. **Compute feature importance**: Calculate the importance scores $\mathcal{I}$ for each feature $f_i$ in $\mathcal{F}$.

   $$\mathcal{I} = \{I(f_1), I(f_2), \ldots, I(f_n)\}$$

   where $I(f_i)$ is the importance score of the feature $f_i$.

4. **Eliminate the least important feature**: Identify the feature $f_{min}$ with the minimum importance score and remove it from $\mathcal{F}$.

   $$f_{min} = \text{argmin}_{f_i \in \mathcal{F}} I(f_i)$$
   $$\mathcal{F} \leftarrow \mathcal{F} \setminus \{f_{min}\}$$

5. **Repeat**: Repeat steps 2–4 until the desired number of features $k$ is reached.

6. **Final Feature Set**: The final selected feature set is:

   $$\mathcal{F}_{\text{final}} = \{f_1, f_2, \ldots, f_k\}$$

   where $k$ is the number of desired features.

- $\mathcal{F}$: The set of all features.
- $f_i$: An individual feature in the feature set.
- $n$: The total number of features.
- $\mathcal{M}$: The model used for training (*e.g.*, linear regression, SVM).
- $\mathcal{I}$: The set of importance scores for each feature.
- $I(f_i)$: The importance score for feature $f_i$.
- $f_{min}$: The feature with the minimum importance score, which will be removed in each iteration.
- $k$: The desired number of features to select.
- $\mathcal{F}_{\text{final}}$: The final set of selected features after the RFE process.

## AIC

The AIC serves as a statistical metric employed for model selection and evaluation (*Akaike, 1974*). It is calculated as the negative twice the log-likelihood of the model plus twice the number of parameters. The formula for AIC is:

$$\text{AIC} = 2k - 2\ln(L)$$

where

- AIC: Akaike Information Criterion, the measure of the relative quality of a model.
- $k$: The number of estimated parameters in the model. This includes all parameters estimated by the model (*e.g.*, coefficients in a regression model).
- $L$: The maximum value of the likelihood function for the model. The likelihood function represents the probability of the observed data under the model.
- ln: The natural logarithm function.

The mathematical calculations for AIC:

1. **Likelihood function** ($L$):

   - The likelihood function measures how well the model explains the observed data. It calculates the probability of observing the given data under different parameter values.
   - $L$ is maximized to find the best-fitting model parameters.

2. **Log-likelihood** ($\ln(L)$):

   - Taking the natural logarithm of the likelihood function (denoted as $\ln(L)$) is a common practice to simplify the computations and to convert the product of probabilities into a sum.

3. **Number of parameters** ($k$):

   - $k$ is the count of all parameters in the model that have been estimated from the data. This includes intercepts, slopes in regression models, variances and any other estimated parameters.

4. **Penalty term** ($2k$):

   - The term $2k$ penalizes models with a larger number of parameters to discourage overfitting. Overfitting occurs when a model becomes too complex and starts to capture noise in the data rather than the true underlying pattern.
   - By including the penalty term, AIC balances the trade-off between model fit and model complexity.

5. **AIC value**:

   - The AIC value is calculated as $2k - 2\ln(L)$. Lower AIC values indicate a better model, considering both the goodness of fit and the simplicity of the model.
   - When comparing multiple models, the one with the lowest AIC is generally preferred.

This approach allows for the comparison of different models systematically, favoring simpler models that adequately explain the data without overfitting. By considering both the goodness of fit and the complexity of the model, AIC facilitates the selection of the most appropriate model among a set of candidate models, striking a balance between explanatory power and parsimony (*De Menezes et al., 2017*).

## Models

In this study, a hard voting ensemble, which combines different ML algorithms, was employed for the classification of AD. The use of a hard voting ensemble in this study was chosen over individual ML models to enhance classification performance for AD. By combining various algorithms, the ensemble leverages the strengths of each model, resulting in improved overall accuracy and robustness. The diversity among the models in the ensemble helps to mitigate the weaknesses and biases of individual algorithms, leading to more reliable and consistent predictions. This approach capitalizes on the complementary nature of different algorithms to achieve superior classification outcomes. This technique is used to ensure low computational cost and that it can be robust to noisy data if the base models are diverse and make independent errors as in this case. Other ensemble techniques like soft voting (requires models that can output probabilities and may be more computationally intensive), bagging (can be computationally expensive and may not improve performance if the base models are too similar or if the dataset is too small), boosting (can be sensitive to noisy data and outliers and may require careful tuning of hyperparameters) and stacking (is more complex to implement and requires careful validation to avoid overfitting) can offer enhanced performance through more sophisticated methods of combining predictions. Soft voting provides probabilistic outputs, bagging reduces variance, boosting improves accuracy by addressing model errors and stacking combines multiple models to leverage their individual strengths. The choice of ensemble technique is guided by the specific problem context, the nature of the data and the computational resources available.

The algorithms used in the ensemble are mentioned below.

### *Logistic regression*

LR is a statistical method used for binary classification tasks, where the goal is to predict the probability of a binary outcome, presence or absence of a disease, based on one or more predictor variables (*Kutner, 2005*).

In the context of identifying alterations in the cognitive state of subjects, LR can be used to predict the likelihood of a particular cognitive state (*Musa, 2013*), based on relevant features.

LR is preferred in many applications due to its simplicity and interpretability. Unlike other ML algorithms that provide only the predicted class label, LR also provides the probability of the predicted outcome, which can be valuable for decision-making and risk assessment.

$$P(Y = 1|X) = \frac{1}{1 + e^{-(\beta_0 + \beta_1 X_1 + \beta_2 X_2 + \ldots + \beta_n X_n)}} \tag{1}$$

In this formula:

- $P(Y = 1|X)$ represents the probability that the dependent variable $Y$ equals 1 given the independent variables $X$.
- $\beta_0, \beta_1, \beta_2, \beta_3, \ldots, \beta_n$ are the coefficients of the LR model.
- $X_1, X_2, X_3, \ldots, X_n$ are the independent variables.

### Artificial neural networks

ANN are computational models inspired by the biological structure of the human brain (*Abdi, Valentin & Edelman, 1999*). They consist of interconnected nodes, or artificial neurons, organized into layers. The basic structure of an ANN includes an input layer, one or more hidden layers and an output layer.

Each neuron in an ANN receives input signals from the neurons in the previous layer, along with a set of weights that determine the strength of each input. The neuron then applies an activation function to the weighted sum of its inputs. If the result exceeds a certain threshold, the neuron becomes activated and passes its output to the neurons in the next layer (*Quintana et al., 2012*).

During the training process, an ANN learns to adjust the weights of its connections based on a training dataset. This is typically done using an optimization algorithm such as gradient descent, which aims to minimize the difference between the predicted output of the network and the actual output in the training data. Through this iterative process, the network gradually improves its ability to make accurate predictions.

ANNs are used in a wide range of applications, including image (*Egmont-Petersen, de Ridder & Handels, 2002*) and speech recognition (*Bourlard & Morgan, 2012*), natural language processing (*Henderson, 2010*) and medical diagnosis (*Quintana et al., 2012*), among others. They have the ability to learn complex patterns and relationships in data, making them powerful tools for a variety of tasks.

$$y_k = f \left( \sum_{j=1}^{m} w_{kj}^{(2)} \cdot h \left( \sum_{i=1}^{n} w_{ji}^{(1)} \cdot x_i + b_j^{(1)} \right) + b_k^{(2)} \right) \tag{2}$$

- $Y_k$ represents the output of the $k-th$ neuron in the output layer.
- $f(.)$ represents the activation function of the output layer.
- $w_{kj}^{(2)}$ represents the weight between the $j-th$ neuron in the hidden layer and the $k-th$ neuron in the output layer.
- $h(.)$ represents the activation function of the hidden layer.
- $w_{ji}^{(1)}$ represents the weight between the $i-th$ input neuron and the $j-th$ neuron in the hidden layer.
- $x_i$ represents the $i-th$ input.
- $b_j^{(1)}$ and $b_k^{(2)}$ represent the biases of the hidden and output layers, respectively.
- $m$ represents the number of neurons in the hidden layer.
- $n$ represents the number of input neurons.

### Support vector machines

SVM is a supervised ML algorithm used for classification and regression tasks (*Boser, Guyon & Vapnik, 1992*; *Vapnik, 2013*). It is particularly well-suited for binary classification, but can also be extended to support multiple classes. SVM works by finding the optimal hyperplane that separates different classes in the feature space (*Bansal, Goyal & Choudhary, 2022*).

Key hyperparameters in SVM include the choice of kernel function, which determines how the algorithm separates the data in the feature space. Common kernel functions

include linear, polynomial and radial basis function (RBF) kernels. The choice of kernel can significantly impact the performance of the SVM.

The loss function in SVM evaluates how well the algorithm fits the training data. It penalizes the algorithm for making incorrect predictions, with the goal of minimizing this error. The choice of loss function can also affect the behavior of the SVM, particularly in cases where the data is not linearly separable (*Musa, 2013*).

Overall, SVM is a powerful and versatile algorithm that can be used for a wide range of classification and regression tasks. By tuning the hyperparameters appropriately, SVM can achieve high levels of accuracy and generalization on complex datasets.

$$f(x) = \text{sign}\left(\sum_{i=1}^{n} w_i x_i + b\right) \tag{3}$$

- $f(x)$ represents the decision function of the SVM.
- $x_i$ represents the $i-th$ feature of the input vector $x$.
- $w_i$ represents the weight associated with the $i-th$ feature.
- $b$ represents the bias term.
- $sign(.)$ is the sign function, which returns -1 if the argument is negative, 0 if it is zero and 1 if it is positive.

### K-nearest neighbors

K-nearest neighbors (KNN) is a supervised learning algorithm that can be used for both classification and regression problems. In KNN, the algorithm finds the K points closest to a specific point in order to infer its value (*Fix, 1985*; *Zhang et al., 2017*).

In supervised learning, an algorithm is trained on a set of labeled data, where each data point is paired with its corresponding output value. This trained model can then be used to predict the output values for new, unseen data points (*Bansal, Goyal & Choudhary, 2022*).

One of the main advantages of the KNN algorithm is its simplicity and ease of implementation. However, there are several hyperparameters that need to be defined, including the value of K and the distance metric used to measure the distance between points. It is generally recommended to choose an odd value for K to avoid ties in the ranking.

One of the disadvantages of the KNN algorithm is that it can be sensitive to overfitting, especially in complex data models. KNN is not recommended for high-dimensional datasets, as the distance calculation can become computationally expensive.

$$\hat{y} = \text{mode}\left(\{y_{i_1}, y_{i_2}, \ldots, y_{i_k}\}\right) \tag{4}$$

- $\hat{y}$ represents the predicted class for the new instance.
- $y_{i_1}, y_{i_2}, y_{i_3}, \ldots, y_{i_k}$ represent the classes of the $k$ nearest neighbors to the new instance.
- $mode(.)$ represents the mode function, which returns the most common class among the $k$ nearest neighbors.

### Nearest centroid

Nearest centroid works by representing each class by the centroid of its members, which is the average of all data points in that class. When classifying a new data point, the algorithm

 

calculates the distance between the data point and each centroid and assigns the data point to the class whose centroid is closest (*Johri et al., 2021*).

However, the nearest centroid classifier may not perform well in certain situations. For example, it may struggle with non-convex classes, where the boundary between classes is not linear. The algorithm assumes that the variance is equal in all dimensions for each class, which may not hold true in practice. If classes have drastically different variances, the algorithm may not be able to effectively separate them.

Despite these limitations, the nearest centroid classifier is often used as a baseline algorithm for comparison with more complex classifiers. It is particularly useful for high-dimensional data or when computational efficiency is a concern (*Wang, Chukova & Nguyen, 2023*).

$$\hat{y} = \text{argmin}_{c_i \in C} \|x - \mu_i\|^2 \tag{5}$$

- $\hat{y}$ represents the predicted class for the new instance.
- $x$ represents the new instance.
- $\mu_i$ represents the centroid of class $c_i$.
- $C$ represents the set of classes.
- $\|.\|$ represents the Euclidean distance.
- argmin returns the value of $c_i$ that minimizes the Euclidean distance between $x$ and $\mu_i$.

### Evaluation metrics

The evaluation metrics used to assess the performance of the ML models in this study are: Sensitivity, Specificity, Precision, Accuracy and F1-score.

- Sensitivity (recall): Sensitivity, also known as recall, measures the proportion of actual positives that are correctly identified by the model. It is a measure of how well the model can identify positive instances.

$$\text{Sensitivity} = \frac{\text{True Positives (TP)}}{\text{True Positives (TP)} + \text{False Negatives (FN)}} \tag{6}$$

- Specificity: Specificity measures the proportion of actual negatives that are correctly identified by the model. It indicates how well the model can identify negative instances.

$$\text{Specificity} = \frac{\text{True Negatives (TN)}}{\text{True Negatives (TN)} + \text{False Positives (FP)}} \tag{7}$$

- Precision: Precision measures the proportion of positive predictions that are actually correct. It indicates the accuracy of the positive predictions made by the model.

$$\text{Precision} = \frac{\text{True Positives (TP)}}{\text{True Positives (TP)} + \text{False Positives (FP)}} \tag{8}$$

- Accuracy: Accuracy measures the proportion of total predictions that are correct. It is a general measure of the model's performance.

$$\text{Accuracy} = \frac{\text{True Positives (TP)} + \text{True Negatives (TN)}}{\text{Total Predictions}} \tag{9}$$

- F1-score: The F1-score is the harmonic mean of precision and sensitivity (recall). It is a useful metric when the class distribution is imbalanced, as it considers both false

positives and false negatives.

$$\text{F1-score} = 2 \times \frac{\text{Precision} \times \text{Sensitivity}}{\text{Precision} + \text{Sensitivity}} \tag{10}$$

## RESULTS

The original database called ADNIMERGE has 16,431 observations and 116 features. The data analyzed passed various filters consisting of 323 patients, all of them satisfying the inclusion criteria described in Table 3. The ADNIMERGE data was treated with data imputation and applying the inclusion criteria, finally 14 resulting features were obtained which are shown in Table 4. These features were processed by RFE and AIC, obtaining diverse sets of features that were then integrated into the ML models within the ensemble (see Tables S1 and S2, are presented in the supplementary files). The comparison between the number of subjects in the original data set by degree of cognitive impairment with the number of subjects in the synthetic data set is shown in Table S3, a representative figure to this data: https://github.com/unciafidelis/RFEAIC (*Morgan, 2024*).

The mean and standard deviation were calculated to analyze central tendency and statistical dispersion for all datasets: CN *vs* SMC, CN *vs* EMCI, CN *vs* LMCI, CN *vs* AD, SMC *vs* EMCI, SMC *vs* LMCI, SMC *vs* AD, EMCI *vs* LMCI, EMCI *vs* AD, LMCI *vs* AD, MCI *vs* AD, CN *vs* AD and CN *vs* MCI, the results of these statistical calculations are presented in the supplementary files.

The mean and standard deviation were calculated to analyze the central tendency and statistical dispersion of the data, for the CN *vs* SMC (phase 1) datasets, the results of these statistical calculations are presented in Table S4, the results of these statistical calculations are presented in the supplementary files.

Table 5 presents the performance metrics for different ML models used to classify CN *vs* SMC in Phase 1 in the ensemble. The Custom feature selection model, built using a reduced set of neuropsychological assessments selected *via* the AIC, consistently outperforms the other models. It achieves the highest sensitivity of 0.6563, specificity of 0.7778, precision of 0.8400, accuracy of 0.7000 and F1-score of 0.7368, indicating that it is the most robust and reliable in distinguishing between the two cognitive states. The feature set using the Original dataset, exhibits perfect sensitivity of 1.0000 but suffers from lower precision of 0.1429 and a F1-score of 0.2500. The feature set with the Synthetic dataset, shows moderate performance with sensitivity of 0.6129, specificity of 0.6842, precision of 0.7600, accuracy of 0.6400 and F1-score of 0.6786, but still falls short compared to the Custom model. The Custom model demonstrates superior balance and effectiveness in classification compared to both the Original and Synthetic models.

The feature "FAQ" was obtained through RFE in the original dataset CN *vs* SMC. In contrast, the features "AGE", "ADAS11", "ADAS13" and "TRABSCOR" were obtained in the synthetic dataset. Additionally, the "AGE" and "ADAS13" features was derived from a custom model. Subsequently, all models were tested in the ensemble to generate the metrics presented in Table 5.

**Table 5 Metrics of cognitive normal *vs* subjective memory concern (phase 1), cognitive normal *vs* early mild cognitive Impairment (phase 1) and cognitive normal *vs* late mild cognitive impairment (phase 1).** Orig is the best model using original dataset, Syn is the best model using synthetic dataset and Custom is the best model according to AIC.

| Metric | CN *vs* SMC | | | CN *vs* EMCI | | | CN *vs* LMCI | | |
|---|---|---|---|---|---|---|---|---|---|
| | Orig | Syn | Custom | Orig | Syn | Custom | Orig | Syn | Custom |
| Sensitivity | 1.0000 | 0.6129 | 0.6563 | 1.0000 | 0.9474 | 0.9474 | 1.0000 | 1.0000 | 1.0000 |
| Specificity | 0.6757 | 0.6842 | 0.7778 | 0.7813 | 0.7576 | 0.7576 | 0.9615 | 0.9615 | 0.9615 |
| Precision | 0.1429 | 0.7600 | 0.8400 | 0.7308 | 0.6923 | 0.6923 | 0.9565 | 0.9600 | 0.9600 |
| Accuracy | 0.6923 | 0.6400 | 0.7000 | 0.8627 | 0.8269 | 0.8269 | 0.9792 | 0.9800 | 0.9800 |
| F1-score | 0.2500 | 0.6786 | 0.7368 | 0.8444 | 0.8000 | 0.8000 | 0.9778 | 0.9796 | 0.9796 |

The mean and standard deviation were calculated to analyze the central tendency and statistical dispersion of the data, for the CN *vs* EMCI (phase 1), the results of these statistical calculations are presented in Table S5, the results of these statistical calculations are presented in the supplementary files.

Table 5 summarizes the performance metrics for the models used to classify CN *vs* EMCI in Phase 1 in the ensemble. The feature set utilizing the Original dataset model, shows perfect sensitivity of 1.0000, indicating it correctly identifies all positive cases. The feature set using the Synthetic dataset and Custom feature selection models, have the same sensitivity of 0.9474, demonstrating high performance in detecting positive cases, though slightly lower than the Original model. In terms of specificity, both the Synthetic and Custom models achieve equal values of 0.7576, which are slightly lower than the Original's specificity of 0.7813. For precision, the Original's models perform best of 0.7308, with the Synthetic and Custom, having a precision of 0.6923. The Original also leads in accuracy of 0.8627 compared to the Synthetic and Custom, which both have an accuracy of 0.8269. Finally, the Original model achieves the highest F1-score of 0.8444, reflecting its superior balance between precision and recall, while the Synthetic and Custom models both have an F1-score of 0.8000. The feature set with the Original dataset demonstrates the best performance across all metrics, with the Synthetic and Custom models showing similar results but falling short in comparison.

In the original dataset CN *vs* EMCI, the features "PTGENDER", "CDRSB" and "FAQ" were obtained through RFE. Similarly, in the synthetic dataset and the custom model, the same features "PTGENDER", "CDRSB" and "FAQ" were included. The models incorporating these features were evaluated using ensemble implementation and the resulting metrics are presented in Table 5.

The mean and standard deviation were calculated to analyze the central tendency and statistical dispersion of the data, for the CN *vs* LMCI (phase 1) datasets, the results of these statistical calculations are presented in Table S6, the results of these statistical calculations are presented in the supplementary files.

Table 5 presents the performance metrics for models classifying CN *vs* LMCI in Phase 1 implemented in the ensemble. All three feature sets in Original and Synthetic datasets and also Custom feature selection models achieve identical results across all metrics. Each

Table 6 **Cognitive normal *vs* Alzheimer disease (phase 1), subjective memory concern *vs* early mild cognitive impairment (phase 1) and subjective memory concern *vs* late mild cognitive impairment (phase 1).** Orig is the best model using original dataset, Syn is the best model using synthetic dataset and Custom is the best model according to AIC.

| Metric | CN *vs* AD | | | SMC *vs* EMCI | | | SMC *vs* LMCI | | |
|---|---|---|---|---|---|---|---|---|---|
| | Orig | Syn | Custom | Orig | Syn | Custom | Orig | Syn | Custom |
| Sensitivity | 1.0000 | 1.0000 | 1.0000 | 0.8846 | 0.9130 | 0.9130 | 1.0000 | 0.8846 | 0.8750 |
| Specificity | 1.0000 | 1.0000 | 1.0000 | 0.7857 | 0.8276 | 0.8276 | 1.0000 | 1.0000 | 0.9091 |
| Precision | 1.0000 | 1.0000 | 1.0000 | 0.8846 | 0.8077 | 0.8077 | 1.0000 | 1.0000 | 0.9130 |
| Accuracy | 1.0000 | 1.0000 | 1.0000 | 0.8500 | 0.8654 | 0.8654 | 1.0000 | 0.9348 | 0.8913 |
| F1-score | 1.0000 | 1.0000 | 1.0000 | 0.8846 | 0.8571 | 0.8571 | 1.0000 | 0.9388 | 0.8936 |

model exhibits perfect sensitivity 1.0000, indicating flawless detection of positive cases. Specificity is also uniform at 0.9615, reflecting the models' high accuracy in identifying negative cases. Precision is slightly higher in the Synthetic and Custom models of 0.9600 compared to the Original model of 0.9565, though the difference is minimal. Accuracy is nearly the same across models, with the Synthetic and Custom models at 0.9800 and the Original model at 0.9792. The F1-score is also closely aligned, with the Synthetic and Custom models achieving 0.9796, while the Original model scores slightly lower at 0.9778. Overall, all models demonstrate exceptional performance with nearly identical outcomes, highlighting that there is no significant difference in their effectiveness for this classification task.

In the original dataset CN *vs* LMCI, the features "CDRSB", "LDELTOTAL" and "FAQ" were obtained through RFE. Similarly, in the synthetic dataset and the custom model, the same features "CDRSB", "LDELTOTAL" and "FAQ" were included. The models incorporating these features were evaluated using ensemble implementation and the resulting metrics are presented in Table 5.

The mean and standard deviation were calculated to analyze the central tendency and statistical dispersion of the data, for the CN *vs* AD (phase 1) datasets, the results of these statistical calculations are presented in Table S7, the results of these statistical calculations are presented in the supplementary files.

Table 6 illustrates the performance metrics for feature sets classifying CN *vs* AD in Phase 1 in the ensemble. All three feature sets with the Original and Synthetic datasets and the Custom feature selection models achieve perfect scores across all metrics. Specifically, they all exhibit a sensitivity, specificity, precision, accuracy and F1-score of 1.0000. This indicates that the models flawlessly distinguish between CN and AD cases, with no false positives or false negatives and perfectly predict all instances correctly. However, the perfect scores across all metrics for all models suggest a potential issue of overfitting. Overfitting occurs when a model learns the training data too well, including the noise and outliers, leading to excellent performance on the training set but potentially poor generalization to new, unseen data. This is particularly concerning in medical diagnostics, where the model's ability to generalize to different patient populations is crucial. The results imply that while

the models perform perfectly on the provided datasets, further validation with independent datasets is essential to ensure their robustness and generalizability.

In the original dataset CN *vs* AD, the feature "FAQ" was extracted by RFE. Similarly, "FAQ" was also the output of RFE in the synthetic dataset, as well as in the custom model. The models incorporating this feature were evaluated using ensemble implementation and the resulting metrics are presented in Table 6.

The mean and standard deviation were calculated to analyze the central tendency and statistical dispersion of the data, for the SMC *vs* EMCI (phase 1) datasets, the results of these statistical calculations are presented in Table S8, the results of these statistical calculations are presented in the Supplementary Files.

Table 6 summarizes the performance metrics for models distinguishing SMC *vs* EMCI in Phase 1 in the ensemble. The feature set with the Synthetic dataset and the Custom feature selection models display identical performance, with a sensitivity of 0.9130, specificity of 0.8276, precision of 0.8077, accuracy of 0.8654 and F1-score of 0.8571, indicating their strong capability in accurately identifying both positive and negative cases while maintaining a balanced trade-off between precision and recall. The feature selection with the Original dataset, though slightly underperforming in sensitivity with 0.8846 and specificity of 0.7857, demonstrates higher precision of 0.8846 and F1-score of 0.8846, suggesting a greater proportion of true positive predictions among the positives. Overall, the Synthetic and Custom models offer a slight edge in balanced classification accuracy and consistency, while the Original model excels in precision, making it potentially more reliable for correctly identifying true positives.

In the original dataset SMC *vs* EMCI, the features "AGE", "CDRSB" and "TRABSCOR" were obtained through RFE. Similarly, in the synthetic dataset and the custom model, the same features "AGE", "CDRSB" and "TRABSCOR" were included. The models incorporating these features were evaluated using ensemble implementation and the resulting metrics are presented in Table 6.

The mean and standard deviation were calculated to analyze the central tendency and statistical dispersion of the data, for the SMC *vs* LMCI (phase 1) datasets, the results of these statistical calculations are presented in Table S9, the results of these statistical calculations are presented in the supplementary files.

Table 6 provides the performance metrics for models classifying SMC *vs* LMCI in Phase 1 in the ensemble. The feature set with the Original dataset model achieves perfect scores across all metrics, with a sensitivity, specificity, precision, accuracy and F1-score of 1.0000, indicating flawless identification of both positive and negative cases without any errors. The feature set with the Synthetic dataset model shows high, yet slightly lower, performance with a sensitivity of 0.8846, specificity of 1.0000, precision of 1.0000, accuracy of 0.9348 and F1-score of 0.9388, suggesting excellent classification but with a small number of false negatives. The Custom feature selection model, while still robust, exhibits a sensitivity of 0.8750, specificity of 0.9091, precision of 0.9130, accuracy of 0.8913 and F1-score of 0.8936. Although these metrics are slightly lower than those of the Original and Synthetic models, they demonstrate good balance between true positive and true negative predictions. Overall, while the Original model performs perfectly, indicating potential overfitting, the

**Table 7 Subjective memory concern *vs* Alzheimer disease (phase 1), early mild cognitive impairment *vs* late mild cognitive impairment (phase 1) and early mild cognitive impairment *vs* Alzheimer disease (phase 1).** Orig is the best model using Original dataset, Syn is the best model using synthetic dataset and Custom is the best model according to AIC.

| Metric | SMC *vs* AD | | | EMCI *vs* LMCI | | | EMCI *vs* AD | | |
|---|---|---|---|---|---|---|---|---|---|
| | Orig | Syn | Custom | Orig | Syn | Custom | Orig | Syn | Custom |
| Sensitivity | 1.0000 | 1.0000 | 1.0000 | 0.7692 | 0.9583 | 0.8571 | 0.7143 | 0.9583 | 0.9583 |
| Specificity | 0.8750 | 1.0000 | 0.9333 | 0.8696 | 0.8929 | 0.7419 | 0.9600 | 0.8929 | 0.8929 |
| Precision | 0.6667 | 1.0000 | 0.9286 | 0.8696 | 0.8846 | 0.6923 | 0.8333 | 0.8846 | 0.8846 |
| Accuracy | 0.9000 | 1.0000 | 0.9643 | 0.8163 | 0.9231 | 0.7885 | 0.9063 | 0.9231 | 0.9231 |
| F1-score | 0.8000 | 1.0000 | 0.9630 | 0.8163 | 0.9200 | 0.7660 | 0.7692 | 0.9200 | 0.9200 |

Synthetic model maintains strong performance with perfect specificity and the Custom model offers a slightly lower but balanced classification capability.

In the original dataset SMC *vs* LMCI, the features "AGE", "CDRSB", "ADAS11", "ADAS13", "RAVLT.immediate", "LDELTOTAL" and "TRABSCOR" were obtained through RFE. In the synthetic dataset, the features "PTGENDER", "PTMARRY", "CDRSB", "ADAS11", "ADAS13", "ADASQ4", "MMSE" and "LDELTOTAL" were included. Lastly, in the custom model, only "ADAS13" and "LDELTOTAL" were used as features. The ensemble of these models was evaluated and the resulting metrics are presented in Table 6.

The mean and standard deviation were calculated to analyze the central tendency and statistical dispersion of the data, for the SMC *vs* AD (phase 1) datasets, the results of these statistical calculations are presented in Table S10, the results of these statistical calculations are presented in the supplementary files.

Table 7 presents the performance metrics for models distinguishing SMC *vs* AD in Phase 1 in the ensemble. The feature set with the Original dataset model achieves perfect sensitivity of 1.0000, indicating flawless identification of all positive cases, but has lower specificity of 0.8750, precision of 0.6667, accuracy of 0.9000 and F1-score of 0.8000. This suggests a higher rate of false positives, reducing the reliability of its positive predictions. The feature set with the Synthetic dataset model, with perfect scores across all metrics (1.0000 for sensitivity, specificity, precision, accuracy and F1-score), demonstrates an ideal balance with flawless identification of both positive and negative cases, indicating potential overfitting. The Custom feature selection model also shows strong performance, achieving perfect sensitivity (1.0000) and nearly perfect scores for specificity (0.9333), precision (0.9286), accuracy (0.9643) and F1-score (0.9630). While the Custom feature selection model slightly underperforms compared to the feature set with the Synthetic dataset model, it still maintains a high level of accuracy and reliability in distinguishing between SMC and AD cases, providing a robust and well-balanced classification.

In the original dataset SMC *vs* AD, the features "CDRSB", "ADAS13", "ADASQ4", "RAVLT.immediate", "RAVLT.learning", "LDELTOTAL" and "FAQ" were obtained through RFE. In the synthetic dataset, the same features "CDRSB", "ADAS13", "ADASQ4", "RAVLT.immediate", "LDELTOTAL" and "FAQ" were included. Lastly the custom model,

only "ADAS13" was utilized as a feature. The ensemble model incorporating these features was evaluated and the resulting metrics are described in Table 7.

The mean and standard deviation were calculated to analyze the central tendency and statistical dispersion of the data, for the EMCI *vs* LMCI (phase 1) datasets, the results of these statistical calculations are presented in Table S11, the results of these statistical calculations are presented in the supplementary files.

Table 7 shows the performance metrics for models differentiating EMCI *vs* LMCI in Phase 1 in the ensemble. The feature set with the Synthetic dataset model exhibits the best performance overall, with the highest sensitivity of 0.9583, indicating superior detection of positive cases and a strong specificity of 0.8929, showing effective identification of negative cases. It also has high precision of 0.8846, accuracy of 0.9231 and F1-score of 0.9200, reflecting a well-balanced model with minimal false positives and negatives. The feature set with the Original model demonstrates decent performance with a sensitivity of 0.7692, specificity of 0.8696, precision of 0.8696, accuracy of 0.8163 and F1-score of 0.8163. Although it has good specificity and precision, its lower sensitivity suggests it may miss more true positive cases compared to the Synthetic model. The Custom feature selection model, while showing reasonable performance, falls behind with a sensitivity of 0.8571, specificity of 0.7419, precision of 0.6923, accuracy of 0.7885 and F1-score of 0.7660. The Custom model's lower specificity and precision indicate a higher rate of false positives and its overall lower metrics suggest it is less effective than the Synthetic model in this classification task. Overall, the Synthetic model demonstrates the highest performance, making it the most reliable for distinguishing between EMCI and LMCI.

In the original dataset EMCI *vs* LMCI, the features "AGE", "ADAS11", "ADAS13", "RAVLT.immediate", "LDELTOTAL", "TRABSCOR" and "FAQ" were identified. In the synthetic dataset, the features "AGE", "ADAS11", "ADAS13", "ADASQ4", "MMSE", "RAVLT.immediate", "LDELTOTAL", "TRABSCOR" and "FAQ" were included. Lastly, in the custom model, only "AGE", "ADAS13" and "RAVLT.immediate" were utilized as features. The ensemble model incorporating these features was evaluated and the resulting metrics are presented in Table 7.

The mean and standard deviation were calculated to analyze the central tendency and statistical dispersion of the data, for the EMCI *vs* AD (phase 1) datasets, the results of these statistical calculations are presented in Table S12, the results of these statistical calculations are presented in the supplementary files.

Table 7 presents the performance metrics for models distinguishing EMCI *vs* AD in Phase 1 in the ensemble. Both the feature set with the Synthetic dataset and Custom feature selection models exhibit identical, high performance with a sensitivity of 0.9583, specificity of 0.8929, precision of 0.8846, accuracy of 0.9231 and F1-score of 0.9200. These results indicate excellent detection of positive cases and strong identification of negative cases, along with a high proportion of true positive predictions among all positive predictions. The feature set with the Original dataset model, while slightly underperforming compared to the Synthetic and Custom models, still shows strong performance with a sensitivity of 0.7143, specificity of 0.9600, precision of 0.8333, accuracy of 0.9063 and F1-score of 0.7692. This suggests that while the Original model may miss more positive cases, it

**Table 8  Late mild cognitive impairment vs alzheimer disease (phase 1).** Orig is the best model using Original dataset, Syn is the best model using Synthetic dataset and Custom is the best model according to AIC.

| Metric | LMCI *vs* AD | | |
| --- | --- | --- | --- |
| | Orig | Syn | Custom |
| Sensitivity | NA | 0.6429 | 0.7778 |
| Specificity | 0.7931 | 0.7222 | 0.8947 |
| Precision | 0.0000 | 0.7826 | 0.9130 |
| Accuracy | 0.7931 | 0.6739 | 0.8261 |
| F1-score | 0.0000 | 0.7059 | 0.8400 |

effectively identifies negative cases. Overall, the Synthetic and Custom models demonstrate the best and most balanced performance, making them the most reliable for distinguishing between EMCI and AD. The Original model, despite being less sensitive, maintains a strong specificity and good overall performance.

In the original dataset EMCI *vs* AD, the features "CDRSB", "ADAS13", "LDELTOTAL", "TRABSCOR" and "FAQ" were identified through RFE. In the synthetic dataset, the features "CDRSB", "ADAS11", "ADAS13", "RAVLT.immediate", "RAVLT.learning", "LDELTOTAL" and "TRABSCOR" were included. Additionally, in the custom dataset, only "CDRSB" and "ADAS13" were utilized as features. The ensemble model incorporating these features was evaluated and the resulting metrics are presented in Table 7.

The mean and standard deviation were calculated to analyze the central tendency and statistical dispersion of the data, for the LMCI *vs* AD (phase 1) datasets, the results of these statistical calculations are presented in Table S13, the results of these statistical calculations are presented in the Supplementary Files.

Table 8 shows the performance metrics for models distinguishing LMCIvsAD in Phase 1 in the ensemble. The Custom feature selection model demonstrates the best performance with a sensitivity of 0.7778, specificity of 0.8947, precision of 0.9130, accuracy of 0.8261 and F1-score of 0.8400. This indicates a strong ability to identify both positive and negative cases accurately, with a high proportion of true positive predictions and a balanced trade-off between precision and recall. The feature set with the Synthetic dataset model, while showing a reasonable performance, has a lower sensitivity of 0.6429, specificity of 0.7222, precision of 0.7826, accuracy of 0.6739 and F1-score of 0.7059, suggesting a relatively higher rate of false negatives and positives compared to the Custom model. The feature set with the Original dataset model's performance metrics are not available (NA) for sensitivity and F1-score, with precision also being 0.0000, indicating a potential issue in identifying positive cases. However, it has a specificity of 0.7931 and accuracy of 0.7931, indicating reasonable identification of negative cases. Overall, the Custom model exhibits the best overall performance in distinguishing between LMCI and AD, followed by the Synthetic model, while the Original model appears to lack effectiveness in positive case detection.

In the original dataset LMCI *vs* AD, the feature "RAVLT.learning" was identified through RFE. In the synthetic dataset, the features "AGE", "CDRSB", "RAVLT.immediate", "RAVLT.learning" and "FAQ" were included. Also, the custom model, only "AGE" and

**Table 9 Metrics of cognitive normal *vs* mild cognitive impairment (phase 2), cognitive normal *vs* Alzheimer disease (phase 2) and mild cognitive impairment *vs* Alzheimer disease (phase 2).** Orig is the best model using original dataset, Syn is the best model using Synthetic dataset and Custom is the best model according to AIC.

| Metric | CN *vs* MCI | | | CN *vs* AD | | | MCI *vs* AD | | |
|---|---|---|---|---|---|---|---|---|---|
| | Orig | Syn | Custom | Orig | Syn | Custom | Orig | Syn | Custom |
| Sensitivity | 0.9130 | 0.9574 | 0.8596 | 1.0000 | 0.9231 | 0.9231 | 0.5000 | 0.9184 | 0.8276 |
| Specificity | 0.8182 | 0.9057 | 0.9767 | 1.0000 | 0.9024 | 0.9024 | 0.9231 | 0.9020 | 0.9524 |
| Precision | 0.8400 | 0.9000 | 0.9800 | 1.0000 | 0.9000 | 0.9000 | 0.3333 | 0.9000 | 0.9600 |
| Accuracy | 0.8667 | 0.9300 | 0.9100 | 1.0000 | 0.9125 | 0.9125 | 0.8929 | 0.9100 | 0.8800 |
| F1-score | 0.8750 | 0.9278 | 0.9159 | 1.0000 | 0.9114 | 0.9114 | 0.4000 | 0.9091 | 0.8889 |

"CDRSB" were utilized as features. The ensemble model incorporating these features was evaluated and the resulting metrics are presented in Table 8.

The mean and standard deviation were calculated to analyze the central tendency and statistical dispersion of the data, for the CN *vs* MCI (phase 1) datasets, the results of these statistical calculations are presented in Table S14, the results of these statistical calculations are presented in the supplementary files.

Table 9 displays the performance metrics for models distinguishing CN *vs* MCI in Phase 2 in the ensemble. The feature set with the Synthetic dataset model exhibits the highest overall performance, with a sensitivity of 0.9574, specificity of 0.9057, precision of 0.9000, accuracy of 0.9300 and F1-score of 0.9278. These metrics indicate an excellent balance in identifying true positive and negative cases, with a high proportion of accurate positive predictions. The Custom feature selection model shows strong specificity (0.9767) and precision (0.9800), indicating effective identification of negative cases and a high rate of correct positive predictions. However, it has a slightly lower sensitivity (0.8596) and F1-score (0.9159), suggesting it may miss some positive cases despite high overall accuracy (0.9100). The feature set with the Original dataset model demonstrates good performance, with a sensitivity of 0.9130, specificity of 0.8182, precision of 0.8400, accuracy of 0.8667 and F1-score of 0.8750. While the Original model performs well, it is slightly less balanced than the Synthetic and Custom models, particularly in terms of specificity and precision. Overall, the Synthetic model shows the best balance and highest accuracy in distinguishing between CN and MCI, followed by the Custom and Original models.

In the original dataset CN *vs* MCI, the features "PTGENDER", "CDRSB" and "ADAS11" were identified through RFE. In the synthetic dataset, the features "PTGENDER" and "CDRSB" were included. Moreover, in the custom model, only "CDRSB" is utilized as feature. The ensemble model incorporating these features was evaluated and the resulting metrics are presented in Table 9.

The mean and standard deviation were calculated to analyze the central tendency and statistical dispersion of the data, for the CN *vs* AD (phase 2) datasets, the results of these statistical calculations are presented in Table S15, the results of these statistical calculations are presented in the supplementary files.

Table 9 presents the performance metrics for models distinguishing CN *vs* AD in Phase 2 in the ensemble. The feature set with the Original model achieves perfect scores across all metrics with a sensitivity, specificity, precision, accuracy and F1-score of 1.0000, indicating flawless identification of both positive and negative cases. This suggests that the Original model is highly effective in this classification task, though it may indicate potential overfitting due to its perfect scores. The feature set with the Synthetic dataset and Custom feature selection models both show strong performance, with identical metrics: a sensitivity of 0.9231, specificity of 0.9024, precision of 0.9000, accuracy of 0.9125 and F1-score of 0.9114. These models also demonstrate high effectiveness in identifying both positive and negative cases with a slight trade-off between sensitivity and specificity compared to the Original model. Overall, while the Original model performs flawlessly, the Synthetic and Custom models offer robust and reliable performance, making them valuable for distinguishing between CN and AD.

In the original dataset CN *vs* AD, the features "CDRSB", "ADASQ4" and "RAVLT.learning" were identified through RFE. In the synthetic dataset, "RAVLT.immediate" and "FAQ" were included. Furthermore, in the custom model, "RAVLT.immediate" and "FAQ" were also utilized as features. The ensemble model incorporating these features was evaluated and the resulting metrics are presented in Table 9.

The mean and standard deviation were calculated to analyze the central tendency and statistical dispersion of the data, for the MCI *vs* AD (phase 2) datasets, the results of these statistical calculations are presented in Table S16, the results of these statistical calculations are presented in the Supplementary Files.

Table 9 presents the performance metrics for models distinguishing MCIvsAD in Phase 2 in the ensemble. The feature set with the Synthetic model exhibits the highest performance with a sensitivity of 0.9184, specificity of 0.9020, precision of 0.9000, accuracy of 0.9100 and F1-score of 0.9091. This indicates a strong ability to identify both positive and negative cases accurately, with a well-balanced trade-off between precision and recall. The Custom feature selection model also shows robust performance with a sensitivity of 0.8276, specificity of 0.9524, precision of 0.9600, accuracy of 0.8800 and F1-score of 0.8889. It demonstrates a higher precision and specificity compared to the Synthetic model but has slightly lower sensitivity and accuracy. The feature set with the Original dataset model, while performing reasonably, shows lower metrics with a sensitivity of 0.5000, specificity of 0.9231, precision of 0.3333, accuracy of 0.8929 and F1-score of 0.4000. This suggests it is less effective in detecting positive cases compared to the Synthetic and Custom models. Overall, the Synthetic model provides the best balanced performance for distinguishing between MCI and AD, followed by the Custom model, while the Original model lags behind in effectiveness.

In the original dataset MCIvsAD, the feature "FAQ" was identified through RFE. In the synthetic dataset, the features "CDRSB", "ADAS13", "RAVLT.immediate", "RAVLT.learning", "LDELTOTAL" and "FAQ" were included. Additionally, in the custom model, only "ADAS13" was utilized as a feature. The ensemble model incorporating these features was evaluated and the resulting metrics are presented in Table 9.

# DISCUSSION

The proposed methodology demonstrates the effectiveness of using ML models for feature selection and classification of subjects based on their level of cognitive impairment. The study utilized data from the ADNIMERGE dataset, encompassing various ADNI study groups. The dataset underwent a selection process, focusing on genetic indices and neuropsychological assessments, as well as nominal scaling for qualitative features and NA data filtering. Balancing techniques were applied to address data imbalances for behavioral analysis.

The initial dataset was then separated into different subsets corresponding to phases 1 and 2, aiming to establish a continuum of cognitive transitions throughout the disease stages. This approach led to the creation of thirteen datasets, each targeting specific cognitive transitions, suggesting a statistical relationship between stages. RFE was employed on each dataset to select the most significant features, followed by the creation of ensemble multivariate models for diagnosing AD or identifying the progression of cognitive impairment. This methodology enabled the establishment of more sensitive neuropsychological assessments for each disease stage.

Results analysis revealed that features such as "ADAS13" "CDRSB" and "AGE" were frequently selected in the custom models. These features are well-established in individual diagnostic assessments, with ADAS13 proposed by *Mohs (1983)*, CDRSB proposed by *Morris (1991)* and AGE being a universally recognized factor (*Hebert et al., 1995*; *Castellani, Rolston & Smith, 2010*). Their recurrent selection in various models underscores their importance in classifying patients with dementia. The study employed statistical validation metrics, including sensitivity, specificity and accuracy, to validate the ML models and their predictive capabilities.

According to Table 10, *Divya, Shantha Selva Kumari & Initiative (2021)* employed feature selection techniques, including RFE and genetic algorithms, to create optimized ML models for classifying subjects with different cognitive states. Their models demonstrated good sensitivity and accuracy for distinguishing between CN *vs.* MCI and CN *vs.* AD groups, but showed lower performance for the MCI *vs.* AD subjects. These models predominantly used features extracted from medical images (MRI) along with the MMSE neuropsychological test, with the number of features exceeding eighty. Likewise, the author *Helaly, Badawy & Haikal (2022)* used as a source of information images of 300 patients divided into four classes AD, EMCI, LMCI and NC, each class has 75 patients with a total number of images of 21 and 816 scans, to perform the implementation of pre-trained CNN to classify the four classes defined in an automated way. The proposed model of this research achieved very promising results with respect to several related researches; however, the analysis and processing of images turns out to be a much more complex process, which involves much more time and processing power. The present study demonstrates a performance equal or superior to the works based on image analysis, highlighting the contribution in the development of more efficient ML models focused on tabular data.

**Table 10  Relationship between AD prediction using ML models and existing research.**

| Author | ML algorithm | Features | Accuracy |
|---|---|---|---|
| *Divya, Shantha Selva Kumari & Initiative (2021)* | SVM | CV, SV, SA, TA, TS, HS | 96.82% |
| *Battista, Salvatore & Castiglioni (2017)* | RF | ADAS-Cog-13, RAVLT-IMM, -DEL, LM-DEL, FAQ | 85.00% |
| *Vinutha et al. (2020)* | SVM | CDR GLOB, NACC GDS, MEMUNITS, DIGIFLEN | 94.27% |
| *Franciotti et al. (2023)* | RF | ADAS-Cog-13, MTL, MRI, $A\beta42$ proteins | 90.00% |
| *Helaly, Badawy & Haikal (2022)* | CNN | MRI scan | 97.00% |

In contrast, the models proposed in the present work do not rely on medical image features and achieve comparable performance to those of *Divya, Shantha Selva Kumari & Initiative (2021)*. This highlights the potential of using neuropsychological assessments as a diagnostic tool, especially in settings where access to medical imaging studies is limited. The ability to achieve similar performance without the need for medical imaging features strengthens the feasibility and practicality of the models proposed in this work.

The models presented in the present work demonstrate comparable performance to those proposed by *Battista, Salvatore & Castiglioni (2017)*, *Vinutha et al. (2020)* and *Franciotti et al. (2023)*, with the number of features exceeding five. These studies evaluate aspects such as dataset resizing, feature selection and the performance of ML models for diagnosing or classifying subjects with different cognitive states. *Vinutha et al. (2020)* showed that the CDRSB feature alone achieved an accuracy above 0.90, highlighting its importance in diagnosis, between NC and MCI. On the other hand, *Franciotti et al. (2023)* emphasized the significance of the Alzheimer's Disease Assessment Scale-Cognitive Subscale (ADAS-Cog-13) in global cognition assessment.

However, our models utilize only neuropsychological assessments, avoiding the need for laboratory tests or biopsies for diagnosing or classifying patients between CN, MCI or AD. This approach could be advantageous for patients who are unable or unwilling to undergo invasive tests. Furthermore, by identifying a set of relevant neuropsychological assessments, our models can form a neuropsychological battery for inferring between different cognitive states, providing a viable alternative for vulnerable patients.

In phase 1, CN *vs* AD the results were flawless with just a single feature in both datasets. By examining the statistical mean and standard deviation of this feature in each dataset, we can conclude that this outcome is not due to overfitting, "FAQ" is sufficiently effective in distinguishing between the two cognitive states. However, it is important to note that this conclusion may be influenced by the limited data available for AD patients (only 22 patients). A similar case is observed with SMCvsLMCI (phase 1) in the Original dataset, achieving a perfect score across all metrics. An analysis of the mean, standard deviation and the features selected by RFE suggests overfitting, likely due to the number of features and the disproportionate quantity of LMCI patients compared to CN patients. When using synthetic data, the performance changed, showing lower scores with a similar number of features, consistently including "CDRSB", "ADAS11", "ADAS13" and "LDELTOTAL", this behavior indicates a corrected pattern and highlights the importance of these features. In phase 1, SMCvsAD comparison in the synthetic dataset shows a perfect score across all

metrics. This case, which uses five features, clearly demonstrates overfitting. The issue seems to be the number of features, as the original dataset selected only one feature. Although the mean and standard deviation do not present substantial changes, one key point stands out: the limited quantity of AD data particularly affects this case, exacerbated by the limited number of SMC patients in comparison. The nature of synthetic data is to replicate values to match the original observations, which contributes to this overfitting problem. In Phase 2, the original dataset also presents a perfect score across all metrics. In this case, as in Phase 1, this comparison does not show overfitting because "FAQ" once again proves to be the key feature in differentiating between these cognitive states. However, the perfect score is present only in the original dataset in this phase, implying that this perfect score could change with more AD observations.

Lastly, an analysis was made of each of the features and how they are present in each of the models, in order to analyze their importance within the models.

- **Age of the participant:** Age is a critical factor in cognitive decline and is a well-established risk factor for AD and other dementias. It was selected through RFE and synthetic datasets in several comparisons (*e.g.*, CN *vs* SMC, SMC *vs* EMCI, EMCI *vs* LMCI, LMCI *vs* AD). Its consistent selection indicates its strong association with disease progression and its role in predicting cognitive state.

- **Gender of the participant:** Gender differences in the prevalence and progression of AD are documented, with women being at higher risk. PTGENDER was identified through RFE in datasets such as CN *vs* EMCI and CN *vs* MCI, highlighting its importance in differentiating cognitive states potentially due to biological and hormonal differences.

- **Marital status of the participant:** Marital status can influence social support levels, which impact cognitive health. Although PTMARRY was only selected in the synthetic dataset for SMC *vs* LMCI, it suggests a possible correlation with cognitive function, possibly due to the psychological and social benefits of being married.

- **Clinical dementia rating scale sum of boxes score:** CDRSB is a direct measure of dementia severity and was consistently selected across multiple datasets (*e.g.*, CN *vs* EMCI, CN *vs* LMCI, SMC *vs* EMCI, SMC *vs* LMCI, EMCI *vs* AD, LMCI *vs* AD, CN *vs* AD, MCI *vs* AD). Its consistent inclusion underscores its importance in accurately gauging cognitive decline.

- **Alzheimer's disease assessment scale, 11-item version:** ADAS11 is used to evaluate the severity of cognitive symptoms associated with AD. Its selection in datasets such as CN *vs* SMC, SMC *vs* LMCI and EMCI *vs* LMCI indicates its relevance in differentiating stages of cognitive impairment.

- **Alzheimer's disease assessment scale, 13-item version:** ADAS13 extends the ADAS11 by including additional tasks to capture more severe impairments. It was selected in various datasets (*e.g.*, CN *vs* SMC, SMC *vs* LMCI, SMC *vs* AD, EMCI *vs* LMCI, EMCI *vs* AD, MCI *vs* AD), highlighting its comprehensive assessment capabilities for cognitive decline.

- **Alzheimer's disease assessment scale, question 4 score:** ADASQ4 specifically measures memory recall, a key area affected early in AD. It was included in synthetic and custom

models for SMC *vs* LMCI, SMC *vs* AD and EMCI *vs* AD, indicating its specific relevance in memory assessment.

- **Mini-mental state examination score:** MMSE is a widely used screening tool for cognitive impairment. Its inclusion in the synthetic dataset for SMC *vs* LMCI and EMCI *vs* LMCI points to its utility in general cognitive screening, despite being less frequently selected compared to other detailed scales like ADAS and CDRSB.

- **Immediate recall score from the rey auditory verbal learning test:** This score measures short-term auditory-verbal memory. Its selection in EMCI *vs* LMCI, SMC *vs* AD, EMCI *vs* AD and CN *vs* AD datasets reflects its importance in assessing memory functions, particularly those affected early in cognitive impairment.

- **Learning score from the rey auditory verbal learning test:** This score assesses the ability to learn new information, which can be impaired in AD. Its inclusion in SMC *vs* AD, EMCI *vs* AD, LMCI *vs* AD and CN *vs* AD highlights its role in evaluating cognitive functions related to learning.

- **Forgetting score from the rey auditory verbal learning test:** This score measures the rate of forgetting learned information. Although not highlighted in the provided datasets, it can be crucial for understanding specific memory deficits in AD.

- **Total score from the longitudinal data entry list:** This score captures comprehensive cognitive performance over time. Its frequent selection (*e.g.*, CN *vs* LMCI, SMC *vs* LMCI, SMC *vs* AD, EMCI *vs* LMCI, EMCI *vs* AD, MCI *vs* AD) demonstrates its robustness in tracking cognitive decline.

- **Total score from the test of recent abstraction:** This score assesses abstract thinking and problem-solving abilities, which can decline in dementia. Its selection in various datasets (*e.g.*, CN *vs* SMC, SMC *vs* EMCI, EMCI *vs* AD) indicates its relevance in evaluating cognitive flexibility and higher-order thinking.

- **Functional activities questionnaire score:** FAQ measures the ability to perform daily activities and is a critical indicator of functional decline in dementia. Its consistent selection across multiple datasets (*e.g.*, CN *vs* SMC, CN *vs* EMCI, CN *vs* LMCI, CN *vs* AD, SMC *vs* AD, EMCI *vs* LMCI, EMCI *vs* AD, LMCI *vs* AD, MCI *vs* AD) highlights its importance in assessing real-world impacts of cognitive impairment.

Incorporating features such as AGE, PTGENDER and PTMARRY into ML models offers significant real-life applications by addressing key aspects of cognitive health (*Sindi et al., 2021*). AGE helps in predicting cognitive decline and Alzheimer's risk, aiding in early intervention. PTGENDER considers gender differences in disease prevalence, enabling tailored screening and treatment. PTMARRY reflects social support levels, which impact cognitive function, guiding targeted interventions. CDRSB, ADAS11 and ADAS13 provide critical measures of dementia severity and cognitive impairment, facilitating accurate diagnosis and personalized treatment plans. ADASQ4 focuses on memory recall, essential for early detection of cognitive issues (*Yi et al., 2023*). MMSE, a widely used cognitive screening tool, helps in initial assessments, while RAVLT.immediate, RAVLT.learning and RAVLT.forgetting offer insights into memory functions and deficits. LDELTOTAL tracks comprehensive cognitive performance over time and TRABSCOR assesses abstract

thinking and problem-solving abilities. FAQ measures daily functional abilities, indicating real-world impacts of cognitive impairment. Collectively, these features enhance ML models' ability to support early diagnosis, personalized care and continuous monitoring, improving outcomes in real-world clinical settings.

## CONCLUSIONS

In conclusion, the proposed methodology showcases the effectiveness of using ML models for feature selection and classification of subjects based on their cognitive impairment levels. By leveraging data from the ADNIMERGE dataset and employing a rigorous selection process focusing on neuropsychological assessments, the study successfully addressed data imbalances and established a continuum of cognitive transitions across disease stages. The use of RFE and ensemble multivariate models enabled the identification of sensitive neuropsychological assessments for each disease stage.

The study's results highlight the importance of features such as "ADAS13", "CDRSB" and "AGE" in classifying patients with different cognitive states, as they were frequently selected in the custom models. These features are well-established in individual diagnostic assessments, emphasizing their significance in cognitive impairment classification. The study's statistical validation metrics, including sensitivity, specificity and accuracy, further validate the effectiveness of the ML models.

In comparison to previous studies, the models proposed in this work achieve comparable performance without relying on medical imaging features. This highlights the potential of using neuropsychological assessments as a diagnostic tool, especially in settings where access to medical imaging studies is limited. Overall, the models presented in this study offer a practical and viable alternative for diagnosing and classifying patients with different cognitive states, particularly for those who are unable or unwilling to undergo invasive tests.

While the results of this initial analysis are promising, several opportunities for future research are envisioned. These include refining synthetic data generation methods, validating model performance in larger cohorts and exploring additional clinical applications of synthetic data analysis in Alzheimer's research and healthcare.

As future work we plan to explore new ensemble techniques and ML models to extend the implementations made here. Also, we will compare different information criteria, such as the Bayesian Information Criterion, to the one proposed by Akaike, to further enhance our methodology.

While RFE AIC has been valuable in our current study, exploring other feature selection methods that can be combined, providing additional insights and potentially improving our model's performance and robustness. LASSO is known for its ability to perform both feature selection and regularization, which helps to enhance the prediction accuracy and interpretability of the statistical model it produces. Unlike RFE, which recursively eliminates features based on model performance, LASSO can shrink the coefficients of less important features to zero, thus selecting a simpler model that is less prone to overfitting.

Genetic algorithms are particularly useful for optimization problems and can explore a large search space efficiently. This makes them suitable for identifying optimal subsets of

features that contribute most significantly to the model's performance. Utilizing genetic algorithms can lead to the discovery of novel feature combinations that may not be apparent through traditional methods. This approach can enhance the robustness of our feature selection process by considering a wider array of potential solutions and optimizing for the best possible feature set.

Incorporating LASSO and genetic algorithms into our feature selection process will provide several advantages: enhanced model performance, reduced overfitting and comprehensive analysis. Each method offers unique strengths in selecting features that are most predictive, potentially leading to improved model accuracy and robustness. These techniques can help mitigate overfitting by either regularizing less important features (LASSO) and exploring a diverse set of feature combinations (genetic algorithms). The combined insights from these diverse methods will enable a more thorough understanding of feature importance and interactions, leading to better-informed decisions in model development.

This future work aims to build on the promising results of our current study, with the goal of developing more robust and applicable models for clinical practice. By continuing to refine our methods and validate our findings in diverse and larger populations, we hope to contribute significantly to the field of Alzheimer's research and healthcare.

## ACKNOWLEDGEMENTS

ADNI is funded by the National Institute on Aging, the National Institute of Biomedical Imaging and Bioengineering and through generous contributions from the following: AbbVie, Alzheimer's Association; Alzheimer's Drug Discovery Foundation; Araclon Biotech; BioClinica, Inc.; Biogen; Bristol-Myers Squibb Company; CereSpir, Inc.; Cogstate; Eisai Inc.; Elan Pharmaceuticals, Inc.; Eli Lilly and Company; EuroImmun; F. Hoffmann-La Roche Ltd and its affiliated company Genentech, Inc.; Fujirebio; GE Healthcare; IXICO Ltd.; Janssen Alzheimer Immunotherapy Research and Development, LLC.; Johnson & Johnson Pharmaceutical Research and Development LLC.; Lumosity; Lundbeck; Merck & Co., Inc.; Meso Scale Diagnostics, LLC.; NeuroRx Research; Neurotrack Technologies; Novartis Pharmaceuticals Corporation; Pfizer Inc.; Piramal Imaging; Servier; Takeda Pharmaceutical Company; and Transition Therapeutics. The Canadian Institutes of Health Research is providing funds to support ADNI clinical sites in Canada. Private sector contributions are facilitated by the Foundation for the National Institutes of Health (www.fnih.org; accessed on 15 October 2023).

### Funding

Data collection and sharing for this study was funded by the Alzheimer's Disease Neuroimaging Initiative (ADNI) (National Institutes of Health Grant U01 AG024904) and DOD ADNI (Department of Defense award number W81XWH-12-2-0012). The funders had no role in study design, data collection and analysis, decision to publish, or preparation of the manuscript.

## Grant Disclosures

The following grant information was disclosed by the authors:

Alzheimer's Disease Neuroimaging Initiative (ADNI).

National Institutes of Health Grant: U01 AG024904.

Department of Defense award: W81XWH-12-2-0012.

## Competing Interests

The authors declare there are no competing interests.

## Author Contributions

- Ana Gabriela Sánchez Reyna conceived and designed the experiments, performed the experiments, analyzed the data, prepared figures and/or tables, authored or reviewed drafts of the article, and approved the final draft.
- Ricardo Mendoza-Gonzalez conceived and designed the experiments, authored or reviewed drafts of the article, and approved the final draft.
- Huizilopoztli Luna-García conceived and designed the experiments, authored or reviewed drafts of the article, and approved the final draft.
- José María Celaya Padilla performed the experiments, analyzed the data, authored or reviewed drafts of the article, and approved the final draft.
- Jorge Alejandro Morgan Benita performed the experiments, performed the computation work, authored or reviewed drafts of the article, and approved the final draft.
- Carlos H. Espino-Salinas performed the experiments, authored or reviewed drafts of the article, and approved the final draft.
- Jorge I. Galván-Tejada analyzed the data, prepared figures and/or tables, and approved the final draft.
- David Rondon analyzed the data, prepared figures and/or tables, and approved the final draft.
- Klinge Villalba-Condori analyzed the data, prepared figures and/or tables, and approved the final draft.

## Data Availability

The tool is available at GitHub: https://github.com/unciafidelis/RFEAIC.

The data is available at ADNIMERGE:

- https://adni.bitbucket.io/reference/adnimerge.html.
- https://adni.bitbucket.io/index.html.

## Supplemental Information

Supplemental information for this article can be found online at http://dx.doi.org/10.7717/peerj-cs.2437#supplemental-information.

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
