# Peer review of "Synthetic data analysis for early detection of Alzheimer progression through machine learning algorithms"

_PeerJ Computer Science, doi:10.7717/peerj-cs.2437_

## Round 0.1 · original submission · Major Revisions

Based on the referee reports, I recommend a major revision of the manuscript. The author should improve the manuscript, taking carefully into account the comments of the reviewers in the reports and resubmit the paper.

·

Basic reporting

Future research directions may focus on refining synthetic data generation methods, validating model performance on larger cohorts, and exploring additional clinical applications of synthetic data analysis in Alzheimer's research and healthcare.

Experimental design

Train each model using both authentic and synthetic datasets separately, considering various hyperparameters and optimization techniques to maximize performance.

Validity of the findings

Validate the reliability of the machine learning models through rigorous evaluation protocols, including hypothesis testing, cross-validation, and sensitivity analysis.

Additional comments

Ensure that the synthetic data generation process accurately captures the underlying distribution and patterns present in authentic patient data without introducing artificial biases or distortions.

Reviewer 2 ·

Basic reporting

-The writing language of the article should be reviewed and errors should be corrected.
- Line 23 in the abstract should be corrected.
- In general, I believe that there is an error in the spelling of the references in the sentence. The complexity should be removed by reviewing the peerj template and editing the references in the text. For example; in the introduction section, such as lines 103, 109, 113...
- Figure 1 in the work of some of the co-authors of the article previously published in mdpi and Figure 1 of this study are visually very similar to each other. It seems that the steps are different. However, I recommend that Figure 1 be changed formally. In this form, it may be thought by readers to be the same.

Experimental design

- Why is the dataset divided into 13 different cases? Does the original data have 13 sections? Or did the authors divide it?
- Why were 2 different phases used?
- What is the termination condition for the Recursive Feature Elimination method?
- On line 519, "these features are shown in Table 3." There is a sentence. However, the feature sets are not included in Table 3.

Validity of the findings

- In the experimental results section, the results are given in Tables 3-15. However, the test results in the tables are not interpreted.
- The features in balanced and unbalanced data sets for 13 different situations and the results obtained with these features should be given in a table instead of being given in separate tables. Additionally, the names of the features in each data set should be added to this table.
- Why was an ensemble method combining different machine learning algorithms used, rather than applying these algorithms separately? Is it thought that the results obtained have improved? If there is improvement, comparative results should be given.

Additional comments

It is necessary to make adjustments to the article, taking into account the deficiencies evaluated under 3 different headings.

Reviewer 3 ·

Basic reporting

The authors should refer to my additional comments.

Experimental design

The authors should refer to my additional comments.

Validity of the findings

The authors should refer to my additional comments.

Additional comments

1. The manuscript would benefit significantly from additional visualizations of both the raw data and the results. This would help to clarify key findings and facilitate a deeper understanding of the data patterns and relationships.

2. It would be advantageous for the authors to conduct a more thorough analysis of the data changes associated with each technique employed. This could include a detailed examination of how and why the data transformations occur and their implications on the results.

3. The discussion of feature selection algorithms, including RFE and AIC, requires further elaboration. The authors should provide more mathematical justification and a comprehensive visual presentation of the data, along with an in-depth analysis of the algorithmic choices and their impact.

4. Table 5 shows all values as 100, which is unusual. The authors should provide a detailed justification for this occurrence, discussing whether it indicates an issue or a positive outcome, and its implications for the study's validity.

5. The presence of perfect scores in other results warrants further investigation. The authors should explore why these scores occurred, evaluate their reliability, and discuss whether they indicate potential problems or benefits.

6. More comprehensive numerical details about the dataset used are necessary. The authors should describe the dataset's characteristics, including its size, distribution, and any preprocessing steps taken.

7. The authors need to provide a clearer rationale for selecting the specific feature selection algorithms used. A comparison with other available techniques and an explanation of why these methods were chosen would enhance the manuscript.

8. The manuscript should include a discussion on the nature of the datasets used, particularly regarding their balance. The authors should explain what steps were taken to address any imbalances and why certain decisions were made.

9. The analysis of the selected features needs to be more comprehensive. The authors should explore why these features were the most effective, why others failed, and how these findings relate to real-world applications. If the model only requires those to perform their task efficiently, then why does the model only require those that were selected to perform the task?

10. The authors should use appropriate data visualizations for the ADNI dataset, comparing the dataset before and after the application of the algorithms. This should be accompanied by a thorough discussion and analysis of the observed changes.

11. The manuscript should include more relevant references to support the claims made. The authors need to critically validate their sources and numerical figures, particularly in the introduction.

12. The literature review in the introduction would benefit from being divided into clear subsections. This would make each discussion more explicit and easier to follow.

13. The authors should provide more information on why they chose hard voting over other ensemble techniques. A comparison with alternative methods and an explanation of the advantages of hard voting in this context would be beneficial.

14. It would be helpful for the authors to compare their results with those of other machine learning algorithms that use different feature selection techniques. They should analyze whether simpler algorithms could achieve comparable or better performance than the chosen ensemble method.

15. The authors should justify why only certain algorithms were selected and consider the inclusion of additional algorithms. This would provide a more comprehensive evaluation of the feature selection techniques.

16. The authors should explore different combinations of the selected algorithms, rather than using them all together using a combination formula. Additionally, considering other ensemble techniques could offer valuable insights into the effectiveness of various methods.
17. The manuscript would benefit from additional proofreading, particularly in improving the clarity and quality of figures to better support the text.

18. At the end of the manuscript, it would be beneficial if a thorough comparison of the results with those of existing studies are presented. This comparison should highlight whether the proposed approach offers significant improvements and provide context for its relative effectiveness.

19. The introduction should include a table that summarizes the pros, cons, gaps, and results of existing studies. This table should be followed by an in-depth analysis and discussion that clarifies how the proposed approach addresses these gaps and is likely to yield better results.

---

## Round 0.2 · accepted · Accept

Author has addressed reviewer comments properly. Thus I recommend publication of the manuscript.

Reviewer 2 ·

Basic reporting

The authors have made all requested changes in the best possible way.

Experimental design

The authors have made the necessary corrections by taking into account the points suggested for this section.

Validity of the findings

The authors have made the necessary corrections by taking into account the points suggested for this section.

Additional comments

The authors have carefully reviewed and implemented all comments. The paper can be accepted as it is.